# Cellular, circuit and transcriptional framework for modulation of itch in the central amygdala

**Vijay K Samineni[1], Jose G Grajales-Reyes[1,2,3], Gary E Grajales-Reyes[4], Eric Tycksen[5], Bryan A Copits[1], Christian Pedersen[6], Edem S Ankudey[1], Julian N Sackey[1], Sienna B Sewell[1], Michael R Bruchas[1,7,8], Robert W Gereau[1,6,8]\***

[1]Washington University Pain Center and Department of Anesthesiology,Washington University School of Medicine, St. Louis, United States; [2]Medical Scientist Training Program, Washington University School of Medicine, St. Louis, United States; [3]Neuroscience Program, Washington University School of Medicine, St. Louis, United States; [4]Department of Pathology & Immunology, Washington UniversitySchool of Medicine, St. Louis, United States; [5]Genome Technology Access Center, Washington University School of Medicine, Seattle, United States; [6]Department of Biomedical Engineering, University of Washington, Seattle, United States; [7]Departments of Anesthesiology and Pharmacology, University of Washington, Seattle, United States; [8]Departmentsof Neuroscience and Biomedical Engineering, Washington UniversitySchool of Medicine, St.Louis, United States

**\*For correspondence:**
gereaur@wustl.edu

**Competing interests:** The authors declare that no competing interests exist.

**Abstract** Itch is an unpleasant sensation that elicits robust scratching and aversive experience. However, the identity of the cells and neural circuits that organize this information remains elusive. Here, we show the necessity and sufficiency of chloroquine-activated neurons in the central amygdala (CeA) for both itch sensation and associated aversion. Further, we show that chloroquine-activated CeA neurons play important roles in itch-related comorbidities, including anxiety-like behaviors, but not in some aversive and appetitive behaviors previously ascribed to CeA neurons. RNA-sequencing of chloroquine-activated CeA neurons identified several differentially expressed genes as well as potential key signaling pathways in regulating pruritis. Finally, viral tracing experiments demonstrate that these neurons send projections to the ventral periaqueductal gray that are critical in modulation of itch. These findings reveal a cellular and circuit signature of CeA neurons orchestrating behavioral and affective responses to pruritus in mice.

## Introduction

As organisms have evolved, it has been essential that they acquire the means to sense physical and chemical threats in the world around them. One such threat detection system is itch, which accompanies unpleasant sensations that evoke strong urges to scratch and promote learned avoidance behavior (*Bautista et al., 2014*; *Han and Dong, 2014*; *Ikoma et al., 2006*; *LaMotte et al., 2014*). Orchestrating adaptive behaviors (e.g., scratching an itch, avoidance of active threats) in the future requires rapid routing of information to brain regions that can encode memories and modify behavior based on prior experiences. The central amygdala (CeA) represents a strong candidate for these functions as the CeA is thought to play a critical role in learning and modifying sensory and emotional memories and translating this information into apt adaptive behaviors (*Fadok et al., 2018*; *Gründemann and Lüthi, 2015*; *LeDoux and Daw, 2018*). Recent studies have implicated the CeA in the regulation of itch (*Albisetti et al., 2019*; *Chen et al., 2016*; *Ehling et al., 2018*; *Mu et al.,*

*2017*), and elevated activity in the CeA has been seen in patients during experimental itch (*Papoiu et al., 2014*; *Vierow et al., 2015*). Nevertheless, it is currently unknown how CeA neurons encode and modify the sensory or emotional components of itch. To address these questions, we used optical imaging, activity-dependent labeling, neural tracing and cell activity-specific RNA-sequencing to systematically investigate chloroquine-activated CeA neurons and their projections in eliciting itch and its related comorbidities.

## Results

We performed fiber photometry recordings from CeA Vgat neurons in awake, behaving mice to assess the activity of CeA neurons in relation to evoked itch/scratch behaviors. To record real-time $Ca^{2+}$ dynamics in the CeA (*Cui et al., 2013*), we expressed the genetically encoded calcium indicator, GCaMP6s, in CeA GABAergic neurons using viral delivery of Cre-dependent GCaMP6s in Vgat-IRES-Cre mice (*Figure 1a*, *Figure 1—figure supplement 1a–c*). As the majority of CeA neurons are GABAergic (*Swanson and Petrovich, 1998*), this approach allows us to target the CeA and avoid picking up photometry signals from neighboring BLA neurons, as could occur if we used non Cre-

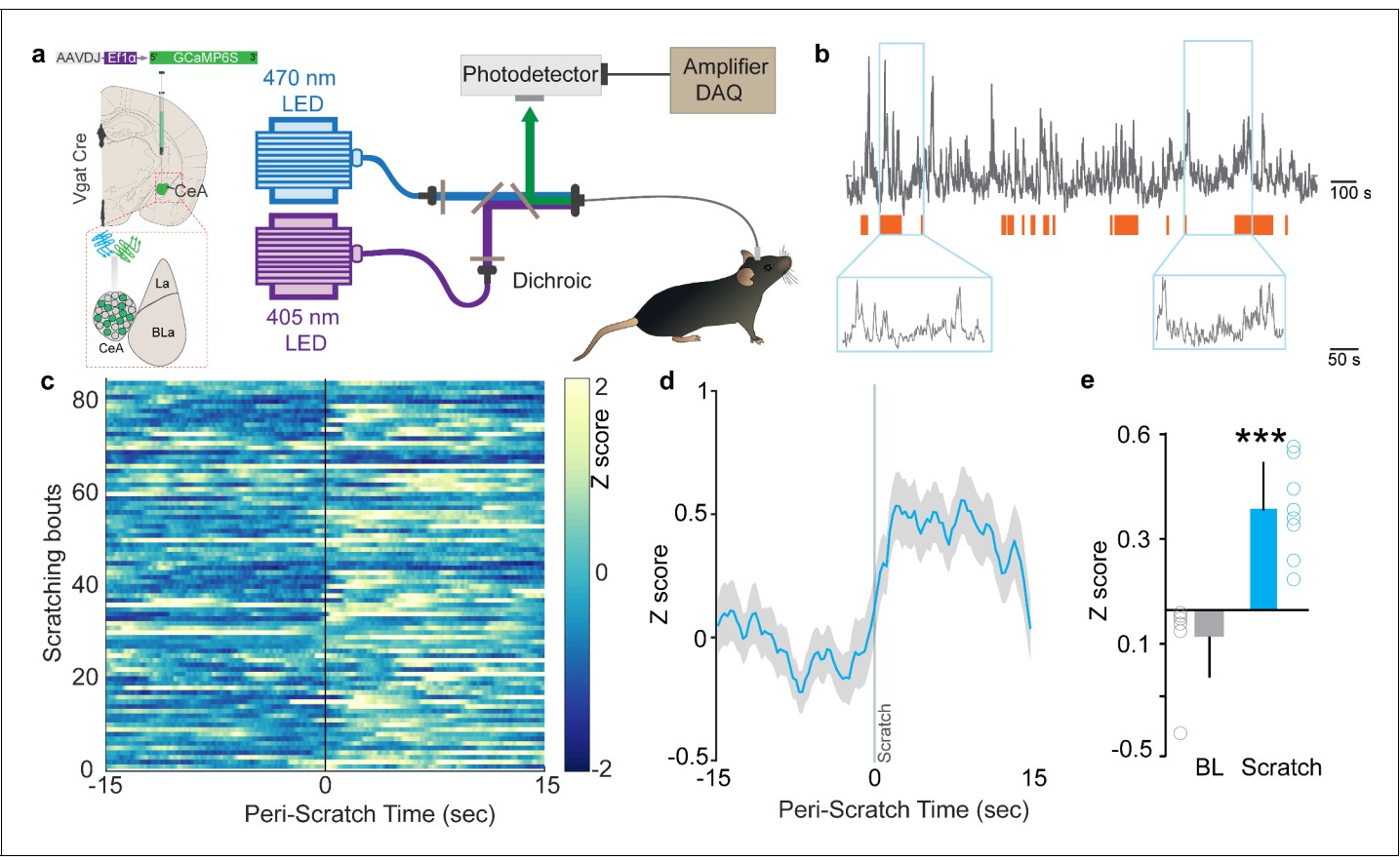

**Figure 1.** Neural dynamics of itch activated with central amygdala (CeA) neurons. (**a**) Scheme demonstrating viral injection strategy and fiber placement to record CeA Vgat neural activity in response to chloroquine. (**b**) Raw Ca2+ dynamics recorded from CeA Vgat neurons and their relationship to chloroquine-evoked scratching bouts (orange bars). (**c**) Heatmap showing Ca2+ dynamics of all trials of Vgat+ve vlPAG neurons relative to the initiation of chloroquine-evoked scratching bouts (time zero). (**d**) Averaged GCaMP6s fluorescence signal of CeA Vgat neurons showing rapid increases in fluorescence on the initiation of scratching bouts. Trace plotted as mean (blue line) ± SEM (gray shading), and the vertical line indicates initiation of scratching bouts. (**e**) Chloroquine-evoked scratching resulted in a significant increase in CeA Vgat neuronal activity as measured by this change in GCaMP6s fluorescence (N = 8, t test, t = 5.923, df = 14, p<0.0001).

The online version of this article includes the following figure supplement(s) for figure 1:

**Figure supplement 1.** Anatomical location of the GCaMP6s-expressing central amygdala (CeA) neurons and fiber placements for imaging activity during itch behaviors.

dependent GCaMP6. Subcutaneous injection of chloroquine in the nape of the neck induced scratching behavior and resulted in robust increases in CeA neuronal activity (*Figure 1b*). This activity commenced with initiation of scratching and stabilized whenever scratching stopped (*Figure 1c–e*), suggesting that the elevated activity was tightly coupled with the act of scratching. Consistent with these real-time dynamic recordings, activity-dependent mapping studies show robust cFos labeling bilaterally in the CeA following chloroquine injection in the nape of the neck compared to saline-injected mice (*Figure 2—figure supplement 1a–c*). We observed no significant differences in cFos labeling between right and left CeA (*Figure 2—figure supplement 1d*).

These observations provide cellular confirmation of prior reports (*Mochizuki et al., 2014*; *Mochizuki et al., 2003*; *Papoiu et al., 2013*) that indicated a possible role for the CeA in itch processing, but the underlying neural circuitry remains to be identified. If CeA neurons function as a key node in the circuit that tightly regulates sensory and affective component of itch, then their activation should trigger potentiation of the itch-scratching cycle and its aversive state. CeA neurons are molecularly heterogeneous and mediate diverse behaviors generally related to negative affect (*John et al., 2015*; *Kalin et al., 2004*; *LeDoux, 2003*; *Ressler and Mayberg, 2007*; *Roozendaal et al., 2009*; *Tye et al., 2011*), so we reasoned global manipulation of CeA neuronal activity would not provide the specificity needed to test the specific roles of chloroquine-activated neurons. To enable the desired selective manipulation of itch-specific neuronal populations in the CeA, we used 'Targeted Recombination in Active Populations' mice (*Guenthner et al., 2013*). These mice express the tamoxifen-dependent CreER$^{T2}$ recombinase from the *Fos* promoter. CreER$^{T2}$ expression is induced in neurons that were recently active. Catalytic activity of CreER$^{T2}$ is stabilized in the presence of 4-hydroxytamoxifen (4-OHT), resulting in transgene recombination. By timing the administration of 4-OHT to coincide with recently increased neuronal activity during acute chloroquine stimuli, we can gain permanent genetic access to chloroquine-responsive CeA neurons (aka FosTRAP mice). To test the validity of this approach, we crossed FosTRAP mice to a Cre-dependent tdTomato flox-stop reporter line (*Madisen et al., 2010*). We injected chloroquine or saline into the nape of the neck, paired with injection of 4-OHT to induce Cre-mediated recombination of the tdTomato in activated (cFos-expressing) neurons (*Figure 2a, b*). FosTRAPing with chloroquine treatment produced robust tdTomato expression in both the right and left CeA (*Figure 2c–e*), and small number of neurons in saline-treated controls (*Figure 2—figure supplement 1e–g*), consistent with our cFos staining results above. This small population of saline TRAPed neurons could be due to the needle stick during the injection itself, and thus could label some pain-responsive CeA neurons. To confirm that the FosTRAPed neurons are specific to the chloroquine-evoked scratching, 1-week post-FosTRAP, we immunostained for c-Fos protein in mice that received an additional chloroquine injection just prior to sacrificing (*Figure 2c, f*). The majority of the tdTomato-positive CeA FosTRAPed neurons faithfully overlap with cFos-positive cells. These results demonstrate that we can efficiently gain genetic access to neurons that are activated by chloroquine.

To test whether reactivating chloroquine-responsive CeA neurons can recapitulate itch behaviors, we expressed the Cre-dependent excitatory opsin, ChR2 (AAV5-EF1a-DIO-ChR2-eYFP), or a control virus (AAV5-EF1a-DIO-eYFP) in the right CeA of FosTRAP mice and FosTRAPed with chloroquine treatment as above (*Figure 2g*). This produces expression of ChR2 specifically in CeA neurons responsive to itch, enabling their selective light-dependent activation. Optogenetic reactivation of FosTRAPed (ChR2$^+$) right CeA neurons resulted in significant spontaneous scratching and grooming behaviors compared to pre-stimulation baseline and photostimulation of eYFP-expressing control mice. Interestingly, although ChR2 was FosTRAPed by injecting chloroquine into the nape of the neck, we observed spontaneous scratching and grooming behaviors directed all over the body (data not shown) in a stimulation frequency-dependent manner (*Figure 2h, i*). Even though some functions of the CeA are lateralized (*Carrasquillo and Gereau, 2007*), elicitation of scratching behaviors is not lateralized to the right CeA as optical stimulation of FosTRAPed ChR2$^+$ neurons in the left CeA also resulted in significant spontaneous scratching behaviors compared to pre-stimulation baseline and photostimulation of eYFP-expressing controls (*Figure 2—figure supplement 2*). This result is consistent with the observation that chloroquine injection induces cFos expression in left and right CeA (*Figure 2—figure supplement 1d*). To further confirm these results and as a complementary approach, we expressed the Cre-dependent excitatory DREADD, hM3Dq (AAV5-hSyn-DIO-hM3Dq-mCh), or a control virus (AAV5-hSyn-DIO-mCh) in the CeA of FosTRAP mice. Chemogenetic

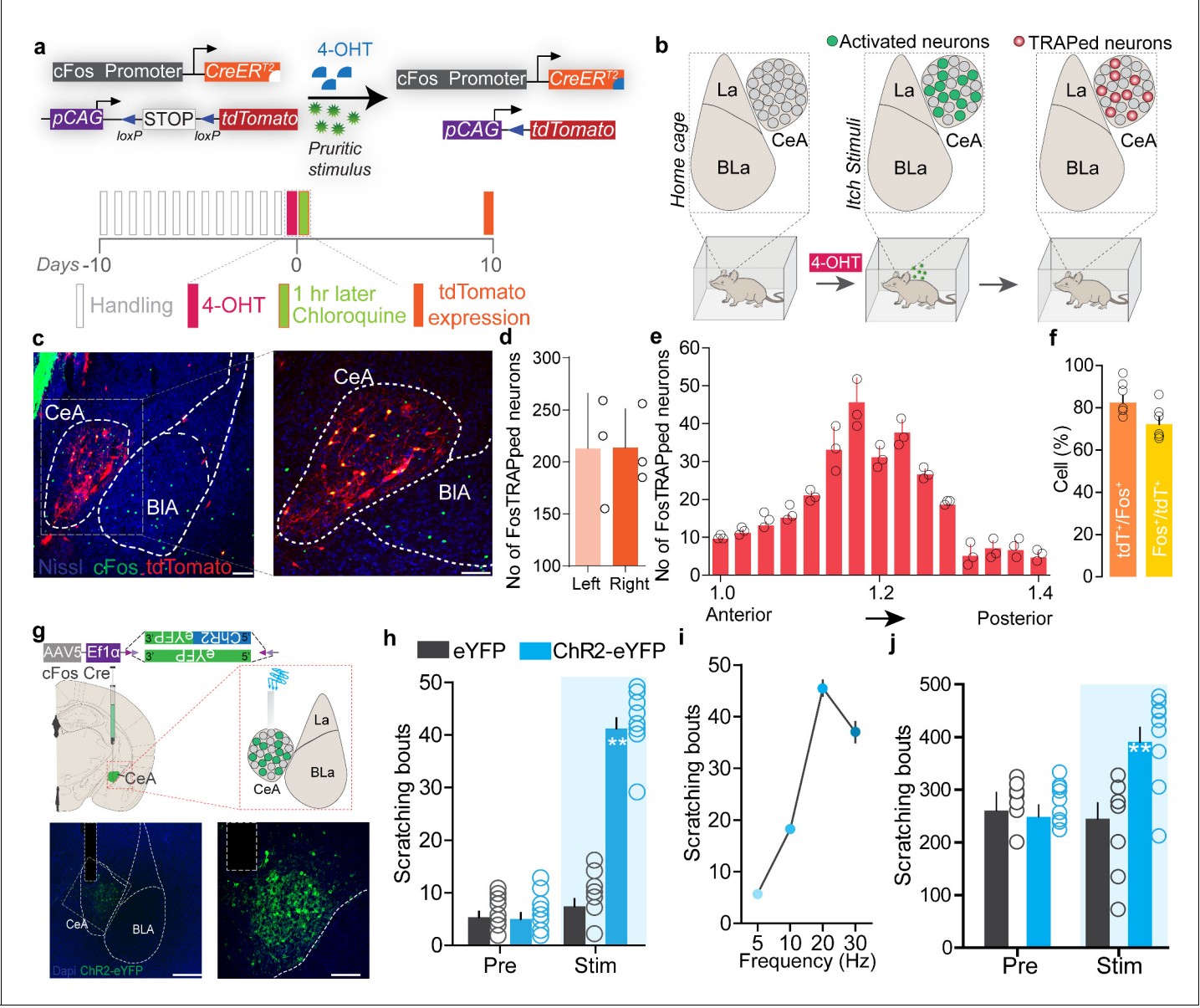

**Figure 2.** Chloroquine-activated central amygdala (CeA) neurons can drive pruritic behaviors. (**a**) FosTRAP strategy to selectively label chloroquine-activated neurons in the CeA. (**b**) Scheme illustrating experimental strategy. (**c**) FosTRAPing with chloroquine-evoked scratching produces robust tdTomato expression in the CeA. Colocalization of chloroquine-TRAPed neurons (red) in the CeA with cFos immunoreactivity (green) following a second administration of chloroquine 7 days post. Scale bar = 85 and 250 μm. (**d**) Quantification of the number of FosTRAPed neurons in left and right CeA after chloroquine injection. n = 3 per group. t test, t = 0.4339, df = 2, p=0.70. (**e**) Rostro-caudal distribution of chloroquine-TRAPed CeA neurons after chloroquine injection. (**f**) Colocalization of chloroquine-activated cFos with tdTomato+ve chloroquine-TRAPed neurons. Relative percentages of Fos+ve neurons that are tdTomato+ve and tdTomato+ve neurons that are Fos+ve. n = 6 per group. t test, t = 2.04 df = 10, p=0.048. (**g**) Scheme to selectively express optogenetic constructs in chloroquine-TRAPed CeA neurons. Illustration and representative section showing fiber optic placement above FosTRAPed CeA neurons expressing ChR2-eYFP (green). Scale bar, 100 μm. (**h**) Photostimulation (20 Hz) of chloroquine-TRAPed CeA neurons produces robust spontaneous scratching. n = 6–11 per group. Pre vs. Stim, F (1,30) = 3; eYFP vs. ChR2, F (1,14) = 3.24, p=0.0001, ANOVA and Bonferroni's for post hoc tests. (**i**) Increases in scratching are frequency dependent. n = 6 per group. (**j**) Optical activation of chloroquine-TRAPed CeA neurons potentiates chloroquine-evoked scratching while no changes were observed in control mice. n = 7 per group. Pre vs. Stim, F (1,12) = 33.15; BL vs. Stim in ChR2, F (1, 6) = 6.915, p=0.0391, ANOVA and Bonferroni's for post hoc tests.

The online version of this article includes the following figure supplement(s) for figure 2:

**Figure supplement 1.** Anatomical location of cFos-expressing (itch-activated) neurons in the central amygdala (CeA) following chloroquine injection in the nape of the neck.

**Figure supplement 2.** Optogenetic reactivation of itch-TRAPed neurons in the left central amygdala (CeA) neurons promotes scratching.

*Figure 2 continued on next page*

*Figure 2 continued*

**Figure supplement 3.** Chemogenetic activation of the central amygdala (CeA) neurons promotes itch behaviors.

**Figure supplement 4.** Chemogenetic manipulation of FosTRAPed central amygdala (CeA) neurons modulates nociceptive behaviors.

activation of FosTRAPed CeA neurons also resulted in significant spontaneous scratching behaviors (*Figure 2—figure supplement 3e*), consistent with the optogenetic results.

In contrast, stimulation of FosTRAPed neurons has no significant effect on hindpaw thermal sensitivity (*Figure 2—figure supplement 4b*) or licking and biting behaviors. However, stimulation of FosTRAPed neurons slightly increased mechanical sensitivity, suggesting that these neurons can encode generalized scratching behavior and hypersensitivity to mechanical stimuli (*Figure 2—figure supplement 4c*). These results suggest that FosTRAPed neurons might be involved in nociceptive processing (*Neugebauer and Li, 2002*). To further determine how reactivation of chloroquine-activated (ChR2[+]) FosTRAPed neurons (hereafter referred to as 'chloroquine-TRAPed neurons') can affect ongoing scratching behaviors, we administered chloroquine and optically activated chloroquine-TRAPed CeA neurons. Chloroquine-evoked scratching was potentiated with optical reactivation of CeA chloroquine-TRAPed neurons while no changes were observed in the eYFP controls (*Figure 2j*). Chemogenetic stimulation of chloroquine-TRAPed neurons produced similar results (*Figure 2—figure supplement 3h*).

Itch is an aversive sensory experience in humans and rodents (*Desbordes et al., 2015*; *Mochizuki et al., 2015*; *Mochizuki et al., 2014*; *Papoiu et al., 2012*; *Papoiu et al., 2013*), and the CeA mediates aversive phenotypes (*Carrasquillo and Gereau, 2007*; *Ciocchi et al., 2010*; *Ehrlich et al., 2009*; *Haubensak et al., 2010*; *Tovote et al., 2016*). Therefore, we wanted to assess whether chloroquine-TRAPed CeA neurons encode negative valence associated with itch. We performed closed-loop real-time place-testing (RTPT) to assess affective state, where an animal freely explores two chambers but receives photostimulation of ChR2[+ve] chloroquine-TRAPed neurons in only one chamber. Reactivation of chloroquine-TRAPed neurons produced robust place aversion to the stimulated side of the chamber while eYFP-FosTRAPed controls did not (*Figure 3b–d*), thus demonstrating that chloroquine-activated CeA neurons carry negative reinforcement signals.

Patients with pruritic skin disorders exhibit heightened anxiety (*Ginsburg, 1995*), and prior studies have shown CeA as a critical hub in coordinating anxiety states (*Ahrens et al., 2018*; *Shackman and Fox, 2016*). Therefore, we evaluated if reactivation of chloroquine-TRAPed CeA neurons can drive anxiety-like behavior using the elevated zero maze (EZM) assay and open-field test (OFT). Optogenetic and chemogenetic reactivation of chloroquine-TRAPed neurons leads to a profound decrease in time spent in the open arms of EZM compared to controls, indicating anxiogenic-like behavioral state (*Figure 3e*, *Figure 3—figure supplement 1b, c*). Reactivation of these neurons also leads to decreased time spent in the center during the OFT, further suggesting that these neurons can drive anxiety-like behavior (*Figure 3—figure supplement 1d–i*). Notably, opto- and chemogenetic reactivation of FosTRAPed neurons did not drive freezing or flight responses in OFT (*Figure 3—figure supplement 1e, g*), suggesting that these neurons are not involved in fear-like behaviors. We used distance and velocity traveled as surrogate measures of freezing and flight behaviors. Although in our experiments assessing itch and pain behaviors we did not observe obvious freezing or flight behaviors, we did not more formally attempted quantify freezing or flight behaviors. Furthermore, stimulation of these neurons also had no effect on feeding and other appetitive behaviors the CeA is reported to evoke (*Douglass et al., 2017*; *Han et al., 2017*; *Kim et al., 2017*; *Li et al., 2017*; *Figure 3—figure supplements 2* and *3*).

Having shown that chloroquine-TRAPed CeA neurons are sufficient to drive itch-related sensory and affective behaviors, we aimed to determine if endogenous activity of these neurons is necessary for itch-related behaviors. We selectively inhibited chloroquine-responsive CeA neurons by expressing the Cre-dependent inhibitory DREADD, hM4Di (AAV5-hSyn-DIO-hM4Di-mCh) or a control virus (AAV5-hSyn-DIO-mCh) in chloroquine-activated CeA neurons of FosTRAP mice (*Figure 4a–c*). Clozapine N-oxide (CNO) application to ex vivo CeA slices from chloroquine-TRAPed mice decreased neuronal excitability to suprathreshold stimuli (*Figure 4d*). Chemogenetic inhibition of CeA chloroquine-TRAPed neurons by CNO injection significantly attenuated chloroquine-evoked scratching compared to pre-CNO baseline and compared to mCherry-expressing controls (*Figure 4e, f*). These

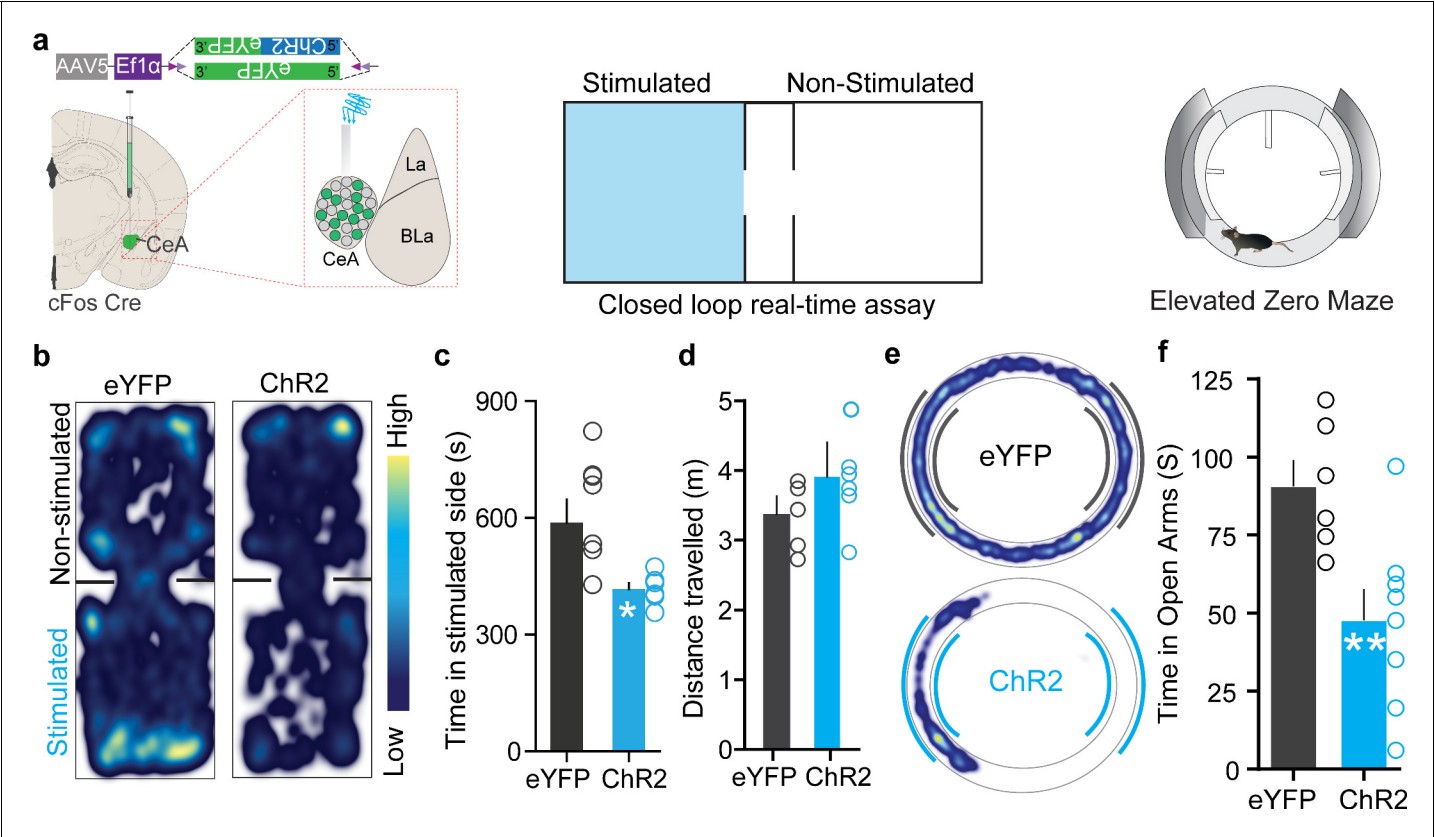

**Figure 3.** Chloroquine-activated central amygdala (CeA) neurons are negatively reinforcing. (**a**) Illustration of strategy to express ChR2/eYFP selectively in chloroquine-TRAPed neurons of the CeA. Experimental schematic of closed loop real-time assay and elevated zero maze (EZM). (**b**) Real-time place aversion assay with spatial location heatmaps of ChR2 and eYFP mice during closed loop optical stimulation. (**c**) Total time spent and (**d**) distance traveled in the photostimulation-paired chamber for ChR2 and eYFP mice. n = 7 per group, t test, t = 2.806, df = 12, p=0.0159, t test, t = 0.7510, df = 12, p=0.4142. (**e**) Representative occupancy heatmap showing spatial location in the EZM of a control mouse (eYFP) and a mouse injected with DIO-ChR2. (**f**) Optogenetic activation of chloroquine-TRAPed CeA neurons causes a significant reduction in time spent in open arms in EZM. Light stimulation was delivered entire time mice were on EZM n = 6–10 per group. t test, t = 5.922, df = 12, p=0.0086.
The online version of this article includes the following figure supplement(s) for figure 3:

**Figure supplement 1.** Optogenetic and chemogenetic activation of FosTRAPed central amygdala (CeA) neurons causes anxiety but no freezing, whereas inhibition of these neurons does not affect anxiety state.
**Figure supplement 2.** Chemogenetic manipulation of FosTRAPed central amygdala (CeA) neurons does not affect feeding behaviors.
**Figure supplement 3.** Chemogenetic manipulation of FosTRAPed central amygdala (CeA) neurons does not affect reward-seeking behaviors.

results suggest that chloroquine-activated (chloroquine-TRAPed) CeA neurons are necessary for chloroquine-induced scratching behavior. We observed no significant effect on thermal or mechanical sensitivity by inhibiting CeA chloroquine-TRAPed neurons (*Figure 2—figure supplement 4*). Furthermore, silencing CeA chloroquine-TRAPed neurons did not lead to freezing or flight responses or anxiolytic effects (*Figure 3—figure supplement 1j–n*). Inhibition of chloroquine-TRAPed neurons also did not affect feeding (*Figure 3—figure supplement 2*) and appetitive behaviors (*Figure 3—figure supplement 3*). Because activating CeA chloroquine-TRAPed neurons lead to robust place avoidance, we hypothesized that inhibiting these neurons would block conditioned place aversion (CPA) to chloroquine. We performed CPA to chloroquine in chloroquine-TRAPed mice expressing either hM4Di or mCherry in chloroquine-responsive neurons (*Figure 4g*). Silencing chloroquine-TRAPed (hM4Di[+ve]) CeA neurons with CNO injection in the chloroquine-paired chamber during conditioning blocked CPA to chloroquine, while CNO-treated mCherry controls exhibited CPA to chloroquine, suggesting that these neurons can robustly modulate the aversive component of itch (*Figure 4h–k*).

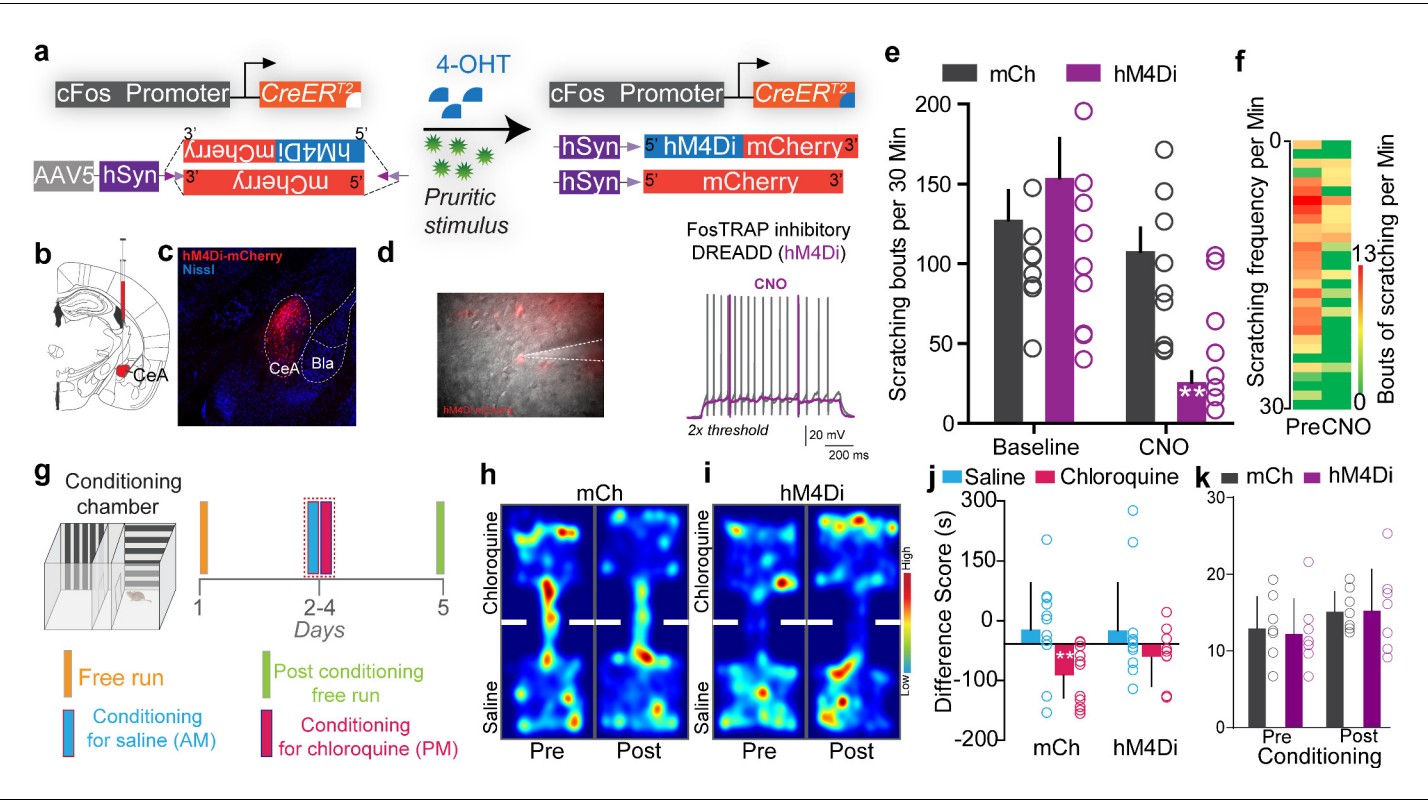

**Figure 4.** Inhibiting chloroquine-activated central amygdala (CeA) neurons impairs aversive learning associated with itch. (**a**) Illustration of strategy to express inhibitory DREADDs selectively in chloroquine-TRAPed neurons of the CeA. (**b**) Experimental timeline to FosTRAP DREADDs in CeA neurons. (**c**) Representative section showing chloroquine-TRAPed CeA neurons expressing hM4Di-mCherry (red). Scale bar, 75 μm. (**d**) Infrared DIC image of CeA chloroquine-TRAPed neurons expressing hM4Di-mCherry. In hM4Di+ve CeA neurons, clozapine N-oxide (CNO) bath application decreased neuronal excitability to suprathreshold stimuli. (**e**) Chemogenetic inhibition of chloroquine-TRAPed CeA neurons leads to a significant reduction in chloroquine-evoked scratching. CNO has no effect on chloroquine-evoked scratching in control mice expressing mCherry. n = 8–9 per group. p=0.0011, ANOVA and Bonferroni's for post hoc tests. (**f**) Heatmap showing averaged chloroquine-evoked scratching bouts pre- and post-CNO in mice expressing hM4Di in CeA. (**g**) Schematic and timeline of conditioned place aversion experimental design with chemogenetic silencing. Representative heatmap showing spatial location of a control mouse injected with the DIO-mCh (**h**) and the DIO-hM4Di DREADD virus (**i**), pre- and post-chloroquine conditioning. (**j**) Change in chamber occupancy time in the chloroquine-paired chamber compared to the saline-paired chamber after chemogenetic silencing. n = 11 per group. p=0.044, ANOVA and Bonferroni's for post hoc tests. (**k**) Distance traveled in chloroquine-paired chamber did not differ pre- and post-conditioning in mCh and hM4Di mice. n = 0.769 per group, ANOVA and Bonferroni's for post hoc tests.

We next sought to understand the circuit context of these chloroquine-activated CeA neurons and explored the downstream nodes that might mediate expression of scratching behaviors. We found that chloroquine-TRAPed CeA neurons send notably dense axonal projections in the ventral periaqueductal gray (vPAG) (*Figure 5a–d*). We also observed projections to the bed nucleus of stria terminalis (BNST), lateral hypothalamus and faint projection in substantia nigra and parabrachial nucleus (PBN). We confirmed vPAG projections by injection CTB into the vPAG (*Figure 5—figure supplement 1a, b*) and also by injecting retro Cre DIO GFP in to the vPAG in Vgat Cre mice (*Figure 5—figure supplement 1c, d*). Injection of RV-GFP into vlPAG of Vgat and Vglut2 Cre mice labeled monosynaptic projections from the CeA consistent with prior work (*Avegno et al., 2018*; *Fadok et al., 2018*; *Haubensak et al., 2010*; *Xu et al., 2016*; *Figure 5—figure supplement 1e–k*). Because the vPAG has previously been shown to contribute to pruritic behaviors (*Gao et al., 2019*; *Samineni et al., 2019*), we focused our functional studies on this CeA→vPAG circuit. If this CeA→vPAG circuit mediates scratching behaviors elicited by the chloroquine-TRAPed CeA neurons, then stimulating this projection should recapitulate these behaviors. As predicted, photostimulating ChR2-expressing chloroquine-TRAPed CeA neuronal terminals in the vPAG recapitulated spontaneous scratching behaviors (*Figure 5e*). Activating chloroquine-TRAPed CeA neuronal terminals in the vPAG did not produce freezing or flight responses. To determine if reactivation of chloroquine-

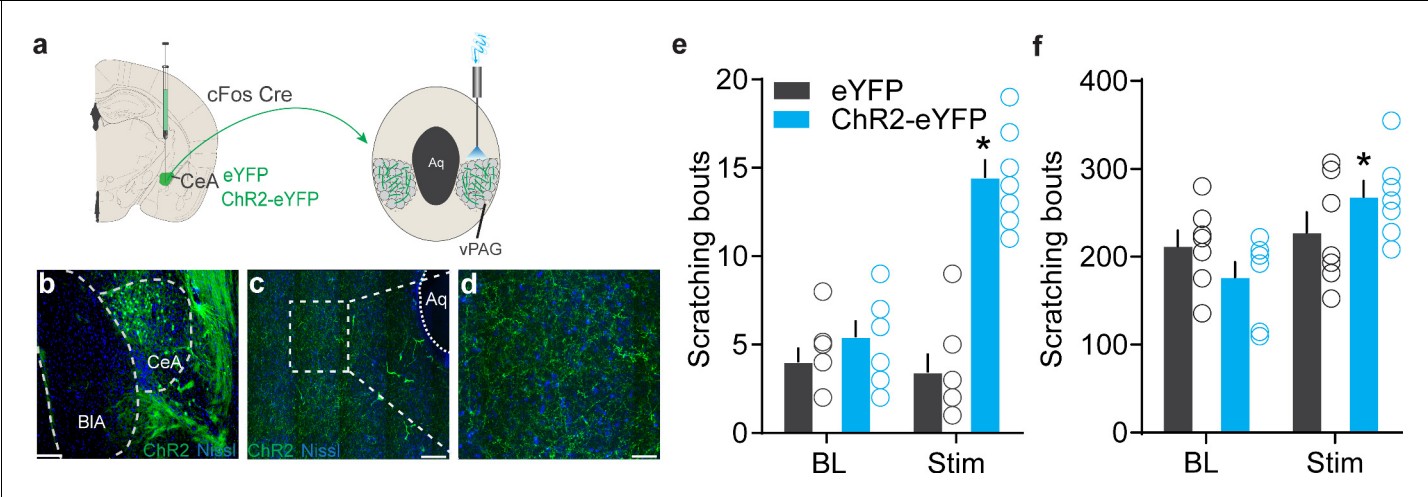

**Figure 5.** Identification of the downstream circuit of chloroquine-activated central amygdala (CeA) neurons. (**a**) Scheme showing expression of ChR2 in chloroquine-TRAPed CeA neurons and their axonal photostimulation in the ventral periaqueductal gray (vPAG). (**b**) FosTRAPed CeA neurons expressing ChR2-eYFP. Scale bar, 125 μm. (**c, d**) Chloroquine-TRAPed ChR2+ve CeA axonal terminals ramify densely in the vPAG. Scale bar, 100 and 25 μm. (**e**) Optogenetic stimulation of FosTRAPed ChR2+ve axonal projections from CeA in the vPAG resulted in significant spontaneous scratching, whereas photostimulation had no effect on scratching in control mice. Pre vs. Stim, F (1,12) = 33.15, p<0.0001, n = 5–7 per group, ANOVA and Bonferroni's for post hoc tests. (**f**) Optical activation of chloroquine-TRAPed CeA neurons potentiates chloroquine-evoked scratching while no changes were observed in control mice. n = 7 per group. BL vs. Stim in eYFP, F (1,6) = 0.019, p=0.8924; BL vs. Stim in ChR2, F (1, 6) = 9.109, p=0.0235, ANOVA and Bonferroni's for post hoc tests.

The online version of this article includes the following figure supplement(s) for figure 5:

**Figure supplement 1.** Anatomical tracing to identify connections between the central amygdala (CeA) and the periaqueductal gray (PAG).

TRAPed CeA→vPAG projections (ChR2$^+$) can influence ongoing chloroquine-evoked scratching behaviors, we administered chloroquine and optically activated chloroquine-TRAPed CeA→vPAG projections. Chloroquine-evoked scratching was potentiated with optical reactivation of CeA→vPAG projections while no changes were observed in the eYFP controls (*Figure 5f*). These results show that the CeA→vPAG neuronal circuit is crucial node in mediating pruritic behaviors.

Lastly, we performed RNA-seq to identify transcriptional profiles of chloroquine-activated CeA cells (*Figure 6a*). To do this, we TRAPed tdTomato in chloroquine-activated CeA neurons as described above and separated the chloroquine-TRAPed neurons from adjacent tdTomato$^{-ve}$ cells for comparative RNA-seq analysis (*Figure 6—figure supplement 1*). Correlation analysis of RNA-seq data revealed chloroquine-TRAPed tdTomato$^{+ve}$ cells and TRAPed tdTomato$^{-ve}$ cells are clustered apart from each other (*Figure 6—figure supplement 2c*). In our sequencing results, we observed that both the tdTomato$^{+ve}$ and tdTomato$^{-ve}$ cells expressed Slc32A1 transcript (VGAT, a marker for GABAergic neurons), consistent with the notion that the majority of CeA neurons are GABAergic. Hierarchical clustering analysis of genes shows highly correlated gene expression patterns that show unique expression profiles in FosTRAPed$^{+ve}$ CeA neurons vs. FosTRAPed$^{-ve}$ CeA neurons (*Figure 6—figure supplement 2d*). We identified numerous highly correlated gene clusters based on their expression levels in FosTRAPed$^{+ve}$ neurons (*Figure 6—figure supplement 2e*). Subsequent analysis of chloroquine-TRAPed neurons revealed significant enrichment of several unique transcripts in the chloroquine-activated neurons (*Figure 6b*). Weighted gene correlation network analysis (WGCNA) of genes identified a cluster of upregulated genes 99% correlated and highly significant for chloroquine-activated neurons (*Figure 6—figure supplement 2f–h*). To link transcriptional profiles of FosTRAPed cells to known CeA functional pathways, we performed pathway analysis. From KEGG and Gene Set Enrichment Analysis (GSEA), we have identified changes in the expression of functionally related candidate genes that are enriched in several pathways (*Figure 6c*). We have identified significantly enriched CeA candidate genes that might be associated with pruritus regulation, as well as significantly genes expressed at significantly lower levels relative to the non-TRAPed cells that could be involved in the suppression of pruritus. To independently confirm our RNA-seq findings, we

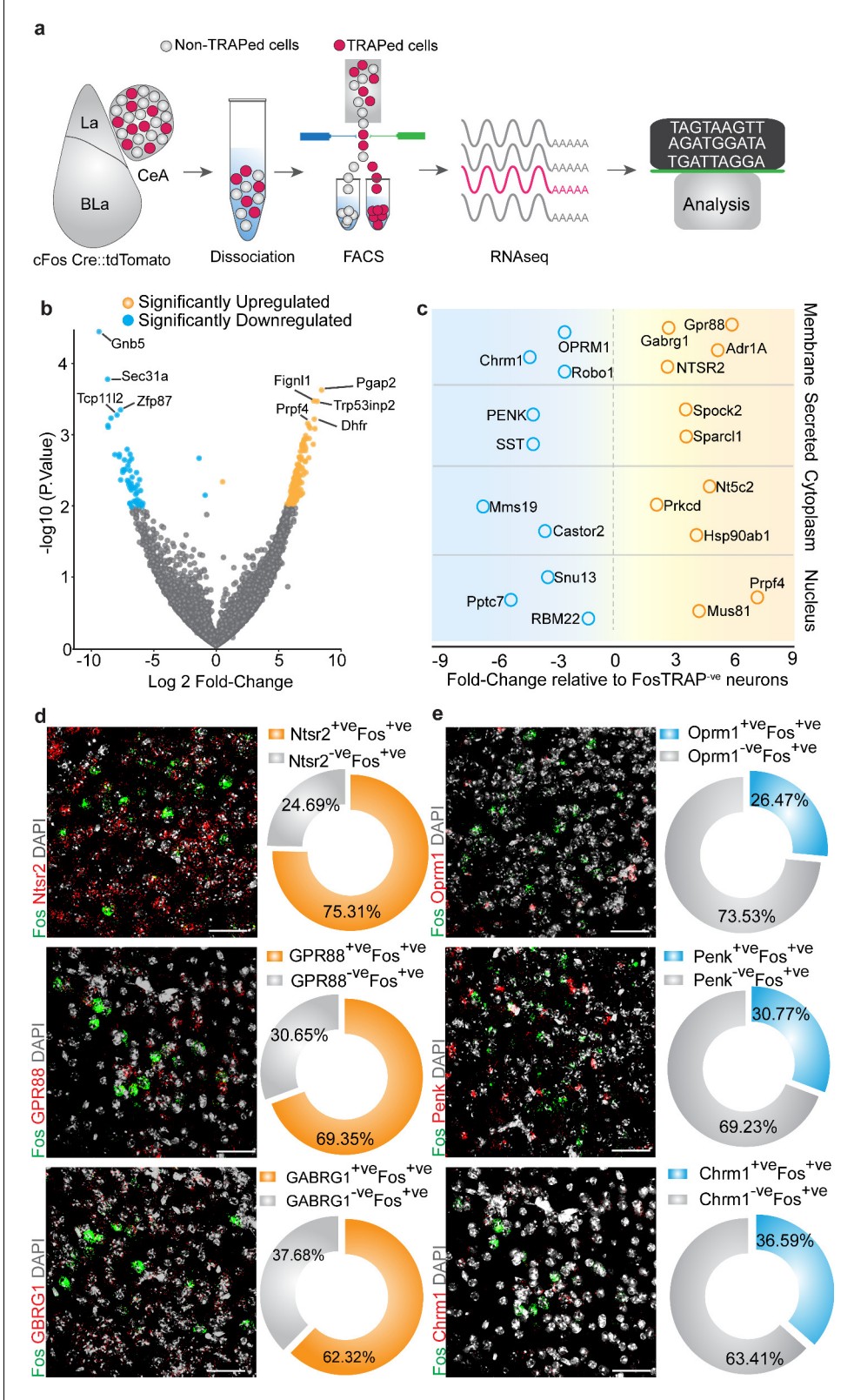

**Figure 6.** Cell-type-specific transcriptomic profiling of chloroquine-activated central amygdala (CeA) cells. (a) Experimental workflow outlining fluorescence-activated cell sorting (FACS) of the FosTRAPed tdTomato+ve and tdTomato-ve CeA neurons for whole-cell transcriptomics analyses. (b) Volcano plot of log2-fold change (x axis) and p values (y axis) showing the transcripts that are differentially expressed in the chloroquine-TRAPed tdTomato+ve CeA cells. Significantly differentially expressed genes are color coded, and genes that have p≤0.001 are indicated on the plot. (c)

*Figure 6 continued on next page*

Figure 6 continued

Candidate genes identified by fold change in expression of genes in significantly enriched KEGG pathways from the FosTRAPed tdTomato+ve CeA cells. (d) Multiplexed fluorescent in situ hybridization (FISH) was used in validating the expression of NTSR2, GPR88 and GABARG1 in itch-activated Fos +ve CeA cells. We observed considerable overlap between NTSR2+ve (75.31% cells), GPR88+ve (69.35%) and Gabrg1+ve (62.32% cells) cells with itch-activated Fos+ve cells in the CeA. (e) Multiplexed FISH was used to verify the overlap of OPRM1, Penk and Chrm1 in itch-activated Fos+ve CeA cells. We find their partial overlap of Fos+ve cells in the CeA with cells that express OPRM1+ve (26.47% cells), Penk+ve (30.77% cells) and Chrm1+ve (36.50% cells). Right corner of each image shows magnification of the inset (yellow box).

The online version of this article includes the following figure supplement(s) for figure 6:

**Figure supplement 1.** Fluorescence-activated cell sorting (FACS) of central amygdala (CeA) FosTRAPed neurons to perform RNA-seq and transcriptional analysis of chloroquine-TRAPed CeA neurons.

**Figure supplement 2.** Transcriptional analysis of central amygdala (CeA) FosTRAPed neurons.

performed dual-color fluorescent in situ hybridization (FISH) to visualize mRNA expression of several candidate genes enriched specifically only in the Fos-positive cells induced by pruritic stimuli. We observed considerable overlap between $Ntsr2^{+ve}$ (75.31% cells per four sections), $Gpr88^{+ve}$ (69.35% cells per four sections) and $Gabrg1^{+ve}$ (62.32% cells per four sections) cells with chloroquine activated Fos$^{+ve}$ cells in the CeA (*Figure 6d*), which were shown to be significantly enriched in chloroquine-TRAPed tdTomato$^{+ve}$ cells. To confirm whether this overlap is specific only to the enriched gene cluster, we also assessed the overlap between chloroquine-activated Fos$^{+ve}$ cells in the CeA and cluster of genes with significantly lower expression in the chloroquine-TRAPed neurons observed from RNA-seq data. We find their partial overlap of Fos$^{+ve}$ cells in the CeA (*Figure 6e*) with cells that express $Oprm1^{+ve}$ (26.47% cells per four sections) $Penk^{+ve}$ (30.77% cells per four sections) and $Chrm1^{+ve}$ (36.50% cells per four sections). These results confirm our findings of differentially expressed genes in chloroquine-activated CeA cells and suggesting that further mining of these sequencing data by the community will reveal important new findings related to chloroquine and its comorbidities.

## Discussion

While there is robust evidence demonstrating the role spinal cord circuits play in driving itch behaviors (*Bautista et al., 2014*; *Han and Dong, 2014*; *LaMotte et al., 2014*; *Ross et al., 2010*; *Sun et al., 2009*), the identity and properties of neural circuits in the brain that coordinate pruritic behaviors are still poorly understood. Neural circuits in the CeA are implicated in pruritic process (*Chen et al., 2016*; *Mu et al., 2017*; *Sanders et al., 2019*), but cells and circuits that can alter pruritic processing in the CeA have been unclear. The CeA is well known to regulate a wide variety of aversive (*Carrasquillo and Gereau, 2007*; *Ciocchi et al., 2010*; *Crock et al., 2012*; *Ehrlich et al., 2009*; *Haubensak et al., 2010*; *Tovote et al., 2016*) and appetitive behaviors (*Cai et al., 2014*; *Carter et al., 2013*; *Douglass et al., 2017*; *Hardaway et al., 2019*; *Kim et al., 2017*; *Robinson et al., 2014*; *Warlow et al., 2020*). Here, we propose a cellular and circuit framework of the CeA in pruritis and its associated affect modulation. By gaining genetic access to neurons that are active specifically during chloroquine-evoked scratching, we were able to selectively identify a diverse repertoire of sensory and aversive behavioral responses mediated by chloroquine-responsive CeA neurons. Activation of chloroquine-responsive neurons in right or left CeA is sufficient to recapitulate spontaneous scratching behaviors, potentiate chloroquine-evoked scratching and produce aversive and anxiety-related behaviors. Furthermore, inhibiting these neurons is sufficient to attenuate ongoing scratching and block its associated aversive component. Our findings reveal the presence of a chloroquine-responsive neuronal population in the CeA, consistent with a recent report (*Sanders et al., 2019*), that is necessary and sufficient to drive itch-related sensory and affective behaviors. We used both optogenetics and chemogenetics in a complementary manner to confirm these results. Conceptually, our findings seem to reconcile prior work (*Chen et al., 2016*; *Mochizuki et al., 2020*; *Papoiu et al., 2014*; *Sanders et al., 2019*; *Sun et al., 2009*; *Vierow et al., 2015*), suggesting a pivotal role of CeA neurons in pruritic processing. The CeA is known to be involved in threat detection and to produce adaptive responses when organisms encounter threatening conditions (*Fadok et al., 2018*; *Gründemann and Lüthi, 2015*; *LeDoux and Daw, 2018*). These

chloroquine-responsive CeA neurons could be a gateway in controlling itch and its associated affective component.

Activation of chloroquine-TRAPed CeA neurons elicits aversive behaviors like place aversion and anxiety-like behaviors, while inhibition of chloroquine-TRAPed CeA neurons produces robust blockade of place aversion to chloroquine. It is well established that the CeA mediates stress-induced anxiety behaviors (*Botta et al., 2015*; *Kalin et al., 2004*; *Li et al., 2017*; *Weera et al., 2020*), but our results show that silencing chloroquine-TRAPed CeA neurons does not have any effect on basal anxiety-like behavior. Future experiments should address whether these neurons are capable of attenuating anxiety during a variety of threats or stressors. Our findings also imply that chloroquine-activated CeA neurons may also be involved to some extent in regulating nociceptive transmission. Chloroquine-activated CeA neurons that drive pruritic processing could represent a subset or overlapping populations that also process nociceptive information. Our additional experiments also suggest that the CeA chloroquine-responsive neurons do not have any effect on feeding, reward-seeking, motor and freezing behaviors. It is possible that these behaviors are driven by distinct molecularly defined cell types in the CeA (*Cai et al., 2014*; *Carter et al., 2013*; *Ciocchi et al., 2010*; *Douglass et al., 2017*; *Ehrlich et al., 2009*; *Hardaway et al., 2019*; *Haubensak et al., 2010*; *Kim et al., 2017*; *Robinson et al., 2014*; *Tovote et al., 2016*; *Warlow et al., 2020*) that are distinct from the chloroquine-activated CeA neurons we have studied here.

We found that chloroquine-activated CeA neurons send functional outputs to the vPAG and activating these CeA→vPAG projections is sufficient to drive scratching. These findings are consistent with recent reports (*Gao et al., 2019*; *Samineni et al., 2019*) showing that activating PAG$^{Vglut2}$ and PAG$^{Tac1}$ neurons produces robust scratching in mice. Based on the results from the present study and reconciling with prior work, a reasonable hypothesis is that information from chloroquine-activated CeA neurons promotes scratching via disinhibition of the PAG$^{Tac1}$ output neurons. As activating these PAG populations does not elicit freezing or escape behaviors, it is possible that PAG neurons that process pruritic information may be distinct from the ones that drive freezing, escape or nociceptive behaviors. Based on our findings, pruritic information arriving in PAG neurons originates at least in part via a CeA chloroquine-responsive neuronal population and contributes to the processing and generation of adaptive responses to pruritis. Our RNA-seq data show that CeA neurons are GABAergic, and anatomical tracing data indicate that CeA neurons that project to the PAG are GABAergic inhibitory populations (*Figure 5—figure supplement 1*). Recent work shows that CeA GABAergic neurons that project to vlPAG can elicit freezing, escape (*Haubensak et al., 2010*; *Tovote et al., 2016*), hunting (*Han et al., 2017*), sleep (*Snow et al., 2017*) and nociception (*Avegno et al., 2018*; *Li and Sheets, 2018*; *Yin et al., 2020*). In our work, we found that the chloroquine-activated CeA→PAG projections drive pruritic behaviors without evoking freezing or escape behaviors, suggesting that these FosTRAPed CeA→PAG projections are distinctly tuned to elicit scratching. The cellular and molecular identities that distinguish these projections from other CeA neurons are yet to be identified (*Steinberg et al., 2020*). A critical task in the future will be to identify and characterize how these different modalities of information are differentially processed via these projections and how the postsynaptic neurons in the PAG differentiate this information to transform it into behavioral output. Here, we establish a critical role for inhibitory projections from CeA to PAG in pruritic regulation. Further studies will be required to more fully understand the mechanisms by which this projection drives itch and associated affective behaviors.

Activating CeA FosTRAP neurons resulted in spontaneous scratching and grooming behaviors directed all over the body, and thus were not restricted to the nape of the neck (where the pruritogen injection was administered for the TRAP). Directed behavior related to sensory information could be organized at the level of sensory and motor cortex. Cortical areas have extensive direct descending projections to the dorsal and ventral horn of the spinal cord (*Joosten et al., 1992*; *Liang et al., 2011*; *Masson et al., 1991*; *Rouiller et al., 1991*). It is not clear how these projections could orchestrate directed scratching behaviors. It is also possible that the CeA is not part of the neural pathways that translate into directed scratching behavior. Neurons in the CeA could receive itch-related information from cortical inputs (*Fadok et al., 2018*), which could be part of the neural pathways that mediate affective aspects like motivation to scratch an itch or suppress itch to evade any immediate potential threat. Our data show that the CeA sends dense projections to the PAG, which in turn modulates spinal pruritic processing via RVM projections, based on previous reports (*Gao et al., 2019*; *Samineni et al., 2019*). There is still much to learn about how this information is

organized. Recent work from *Gao et al., 2019*, suggests that activating Tac1 neurons can drive robust scratching behaviors; this suggests that there could be parallel circuits downstream of the CeA that can evoke and inhibit itch-evoked scratching.

Recent work in the VTA shows that pruritogen-evoked scratching elevates the activity of dopamine neurons, and this elevated activity is required for the hedonic aspects of pruritogen-evoked scratching (*Su et al., 2019*; *Yuan et al., 2018*). The VTA is known to send projections to the CeA. It is possible that these projections encode aversive aspects that we have seen in the CeA (*Leshan et al., 2010*; *Zhou et al., 2019*). What we cannot parse from our data is whether neurons that encode itch evoked by pruritogens and those that respond to scratching are the same or distinct CeA sub populations. It is possible that there are multiple populations driving sensory and motor aspects of itch-scratch behaviors. We also observed projections from the TRAPed neurons to the BNST, lateral hypothalamus and faint projection in substantia nigra and PBN in addition to the vPAG projections. It is possible that these downstream regions could also play a critical role in different aspects of pruritis. There is now literature suggesting that scratching on the site of pruritogen application can suppress neural activity in spinothalamic neurons (*Davidson et al., 2009*). In these neurons, activity elicited by pruritogens can be completely abolished by scratching, suggesting that relief of itch by scratching results from suppression of activity in the spinal cord. There are additional studies now showing that supraspinal projections from the PAG and RVM directly modulate this activity in a state-dependent manner (*Gao et al., 2019*; *Samineni et al., 2019*). Active inhibition of scratching could take place downstream of CeA when PAG and RVM neurons are engaged in inhibiting ongoing spinal pruritic transmission.

The CeA a is highly molecularly heterogeneous region that is known to express a diverse array of neuropeptides, receptors and cellular machinery that are unique to CeA in integrating and orchestrating neuromodulatory functions (*Kim et al., 2017*; *Zirlinger and Anderson, 2003*; *Zirlinger et al., 2001*). To understand the unique genetic identity of CeA neurons that regulate pruritic behaviors, we performed RNA-seq of chloroquine-activated CeA neurons by TRAPing these neurons with pruritic stimuli. These activity-dependent RNA-Seq data show extensive molecular programs that are selectively enriched in the chloroquine-responsive neurons relative to other cells in CeA. We observe significant enrichment of NTSR2, GPR88 and Gabrg1 in chloroquine-activated neurons, and relatively lower expression of OPRM1, PENK and Chrm1, suggesting that these genes could be critical candidates in regulating pruritis and its associated anxiety. One other interesting observation from our sequencing dataset is the relative enrichment of Prkcd transcript in FosTRAP +ve vs. FosTRAP-ve cells. Prkcd+ve cells in CeA have been shown to be involved in fear and pain processing (*Cai et al., 2014*; *Haubensak et al., 2010*; *Wilson et al., 2019*). It would be interesting to see what role these cells play in pruritic behaviors. It is also possible that the needle stick associated with 4-OHT injection could label a small population of CeA neurons involved in fear or pain processing, and this could impact our sequencing dataset to some extent. In our analysis, we have followed up on six candidate genes, but the dataset we have generated here will be an immensely valuable resource to the neuroscience community interested in the role of CeA neurons in modulation of sensory and affective behaviors.

# Materials and methods

**Key resources table**

| Reagent type (species) or resource | Designation | Source or reference | Identifiers | Additional information |
|---|---|---|---|---|
| Species (*Mus musculus*), strain | *Ai9-tdTomato mice (B6.Cg-Gt(ROSA)26Sortm9(CAG-tdTomato)Hze/J)* | The Jackson Laboratory | 007909 | Ai9 |
| Species (*Mus musculus*), strain | *FosCreERT2 mice (B6.129(Cg)-Fostm1.1(cre/ERT2)Luo/J)* | The Jackson Laboratory | 21882 | FosCreER |
| Species (*Mus musculus*), strain | Vgat-ires-Cre (*Slc32a1$^{tm2Lowl}$*) | The Jackson Laboratory | 028862 | Vgat Cre |

*Continued on next page*

*Continued*

| Reagent type (species) or resource | Designation | Source or reference | Identifiers | Additional information |
|---|---|---|---|---|
| Species (*Mus musculus*), strain | Vglut2-ires-Cre (*Slc17a6*<sup>tm2</sup>) | The Jackson Laboratory | 028863 | Vglut2 Cre |
| Species (*Mus musculus*), strain | C57BL\6J | In bred | NA | NA |
| Recombinant DNA reagent | rAAV5/hSyn-DIO-hM3Dq-mCherry | University of North Carolina Vector Core | NA | 75 nL of virus |
| Recombinant DNA reagent | rAAV5/hSyn-DIO-hM4Di-mCherry | University of North Carolina Vector Core | NA | 75 nL of virus |
| Recombinant DNA reagent | rAAV5-DIO-ChR2-eYFP | University of North Carolina Vector Core | NA | 100 nL of virus |
| Recombinant DNA reagent | rAAV5/hSyn-DIO-mCherry | University of North Carolina Vector Core | NA | 75 nL of virus |
| Recombinant DNA reagent | rAAV5-DIO-eYFP | University of North Carolina Vector Core | NA | 100 nL of virus |
| Recombinant DNA reagent | rAAV5/EF1α-FLEX-TVAmCherry | University of North Carolina Vector Core | NA | 75 nL of virus (1:1 with RG) |
| Recombinant DNA reagent | rAAV5/CAG-FLEX-RG | University of North Carolina Vector Core | NA | 75 nL of virus (1:1 with TVA) |
| Recombinant DNA reagent | EnvA G-deleted Rabies-GFP | University of North Carolina Vector Core | NA | 100 nL of virus |
| Chemical compound, drug | Clozapine-N-oxide (CNO) | Enzo Life Sciences | BML-NS105 | NA |
| Chemical compound, drug | 4-Hydroxytamoxifen | Sigma–Aldrich | H6278-10MG | NA |
| Chemical compound, drug | Chloroquine | Sigma–Aldrich | C6628 | NA |
| Antibody | Rb-mCherry | Clontech | Cat. #: 632543 | 1:1000, RRID:AB_2307319 |
| Antibody | Ch-GFP | AVES | A11122 | 1:2000, AB_10000240 |
| Antibody | Rb-cFos | Cell Signaling | Cat. #: D82C12 | 1:1000, RRID:AB_10557109 |
| Sequence-based reagent (smFISH) | mm-Fos | Advanced Cell Diagnostics | 316921 | NA |
| Sequence-based reagent (smFISH) | mm- Ntsr2 | Advanced Cell Diagnostics | 452311 | NA |
| Sequence-based reagent (smFISH) | mm- GPR88 | Advanced Cell Diagnostics | 317451 | NA |
| Sequence-based reagent (smFISH) | mm-Penk | Advanced Cell Diagnostics | 318761 | NA |
| Sequence-based reagent (smFISH) | mm-Gabarg1 | Advanced Cell Diagnostics | 501401 | NA |
| Sequence-based reagent (smFISH) | mm-Oprm1 | Advanced Cell Diagnostics | 315841 | NA |
| Sequence-based reagent (smFISH) | mm-Chrm1 | Advanced Cell Diagnostics | 495291 | NA |
| Software, algorithm | Ethovision XT | Noldus | https://www.noldus.com/ethovision-xt | NA |
| Software, algorithm | Prism7 | GraphPad | https://identifiers.org/RRID/RRID:SCR_002798 | NA |
| Software, algorithm | MATLAB, 2018b | MathWorks | https://www.mathworks.com/products/matlab.html | NA |

*Continued on next page*

*Continued*

| Reagent type (species) or resource | Designation | Source or reference | Identifiers | Additional information |
|---|---|---|---|---|
| Software, algorithm | RZ5P | Tucker-Davis Technologies | https://www.tdt.com/system/fiber-photometry-system/ | NA |

## Animals

All experiments were conducted in accordance with the National Institute of Health guidelines and with approval from the Animal Care and Use Committee of Washington University School of Medicine (approved protocol 20-0078). Mice were housed on a 12 hr light-dark cycle (6:00 am to 6:00 pm) and were allowed free access to food and water. All animals were bred onto C57BL/6J background, and no more than five animals were housed per cage. Male littermates between 8 and 12 weeks old were used for experiments. We conducted a pilot experiment using both male and female mice. We did not observe any differences between groups, and thus did not account for sex differences in our power analysis when we designed the comprehensive study. As this is a resource-intensive study, we proceeded to focus the full study on a single sex, and in this case we used only male mice. *FosCreERT2 mice (B6.129(Cg)-Fostm1.1(cre/ERT2)Luo/J)*; *stock #21882, Ai9-tdTomato mice (B6.Cg-Gt(ROSA)26Sortm9(CAG-tdTomato)Hze/J)*; *stock #007909,* Vgat-ires-Cre (*Slc32a1$^{tm2Lowl}$*; *stock #028862.*), Vglut2-ires-Cre (*Slc17a6$^{tm2Lowl}$*; *stock # 028863*) and C57BL\6J mice were purchased from Jackson Laboratories and colonies were established in our facilities. For all the behavioral experiments, heterozygous cFos-Cre male mice were used, and for Cfos co-staining and sequencing experiments were performed on heterozygous cFos-Cre male mice crossed to homozygous Ai9mice from Jackson Laboratory. Litters and animals were randomized at the time of assigning experimental conditions for the whole study. Experimenters were blind to treatment and genotype.

## Viral constructs

Purified and concentrated adeno-associated viruses coding for Cre-dependent hM3Dq-mCherry (rAAV5/hSyn-DIO-hM3Dq-mCherry; $6 \times 10^{12}$ particles/mL, lot number: AV4495c and lot date: 02/23/2012) and hM4D-mCherry (rAAV5/hSyn-DIO-hM4Di-mCherry; $6 \times 10^{12}$ particles/mL, lot number: AV4496c and lot date: 11/20/2012), control mCherry (rAAV5/hSyn-DIO-mCherry; $3.4 \times 10^{12}$ particles/mL, lot number: AV5360 and lot date: 04/09/2015), ChR2- eYFP (rAAV5-DIO-ChR2-eYFP; $4.8 \times 10^{12}$ particles/mL, lot number: AV4313Y and lot date: 04/21/2017) and control eYFP (rAAV5-DIO-eYFP; $3.3 \times 10^{12}$ particles/mL, lot number: AV4310i and lot date: 07/21/2016) were used to express in the FosCreERT2 mice. Helper virus, AAV1-EF1α-FLEX-TVAmCherry (rAAV5/EF1α-FLEX-TVAmCherry; $4 \times 10^{12}$ particles/mL) and AAV1-CAG-FLEX-RG (rAAV5/CAG FLEX-RG; $3 \times 10^{12}$ particles/mL) were mixed at a ratio of 1:3 and then injected into the vPAG. Three weeks later, EnvA G-deleted Rabies-GFP ($3.9 \times 10^9$ particles/mL) was injected in the vPAG. All vectors except rabies virus were packaged by the University of North Carolina Vector Core Facility. Rabies virus was purchased from Salk Gene Transfer, Targeting and Therapeutics Core. All vectors were aliquoted and stored in −80°C until use.

## Stereotaxic surgeries

Mice were anesthetized with 1.5–2.0% isoflurane in an induction chamber using isoflurane/breathing air mix. Once deeply anesthetized, mice were secured in a stereotactic frame (David Kopf Instruments, Tujunga, CA) where surgical anesthesia was maintained using 2% isoflurane. Mice were kept on a heating pad for the duration of the procedure. Preoperative care included application of sterile eye ointment for lubrication, administration of 1 mL of subcutaneous saline and surgery-site sterilization with iodine solution. A small midline dorsal incision was performed to expose the skull and viral injections were performed using the following coordinates: CeA, −1.24 mm from bregma,±2.8 mm lateral from midline and 4.5 mm ventral to skull. Viruses were delivered using a stereotaxic-mounted syringe pump (Micro4 Microsyringe Pump Controller from World Precision Instruments) and a 2.0 µL Hamilton syringe. Injections of 75–100 nL of the desired viral vectors into the area of interest were performed at a rate of 1 µL per 10 min. We allowed for a 10 min period post injection for bolus

diffusion before removing the injection needle. Postoperative care included closure of the cranial incision with sutures and veterinary tissue adhesive, and application of topical triple antibiotic ointment to the incision site. Animals were monitored while on a heating pad until they full recovery from the anesthetic.

## Cannula implantation

The surgical protocol was the same as described above for viral injections. Fiber optic implants were fabricated using zirconia ferrules (Thorlabs) and from 100 µm diameter fiber (0.22 numerical aperture [NA], Thorlabs). Fiber optic cannulas (length 5 mm) were implanted at the CeA and the PAG and fixed to the skull using two bone screws (CMA anchor screws, #7431021) and dental cement. The following coordinates were used for implantation: CeA, −1.24 mm from bregma, ±2.8 mm lateral from midline and 4.25 mm ventral to skull and the PAG, −4.84 mm from bregma, ±0.5 mm lateral from midline and 2.7 mm ventral to skull. Mice were allowed to recover for 14 days before behavioral analysis. Animals in which cannulas placement missed the CeA or vlPAG target were excluded from the study.

## Chemogenetic manipulation

For chemogenetic control of CeA FosTRAPed neurons, cFos-Cre mice were injected with Cre-dependent control mCh, hM3Di or hM4Dq viruses. DREADD constructs used in this study were validated previously in our lab for their functional expression in the PAG, including their ability to increase (hM3Dq) or decrease (hM4Di) neuronal firing in slices from animals expressing these viral constructs (*Samineni et al., 2017a*). Three weeks later, mice were injected with 4-OHT to express Cre-dependent DREADDs, CNO (BML-NS105 from Enzo Life Sciences) was injected 30 min before doing behavioral experiments and data were collected between 30 min and 2 hr post-injection. All baselines for pruritic responses were recorded 3 weeks after the FosTRAP and 1 week prior to the CNO administration. We used 5 mg/kg CNO as a dose of CNO and showed no signs of behavioral changes in control vector-expressing animals.

## Optogenetic manipulations

For all the behavioral experiments, mice were acclimated to tethered fibers for 5 days before initiation of the experiments. Mice were habituated to tethering with lightweight patch cables (components: Doric Lenses) that are connected to a laser (Shanghai laser, 475 nm). To prevent impediment of movement from the tethered cables, we coupled patch cables to an optical commutator (Doric Lenses). An arduino was programed and connected to the laser to deliver 5, 10, 20 and 30 Hz (5 ms width, 10 mW/mm$^2$) photostimulation in FosCre mice.

## Activity-dependent FosTRAP labeling

### 4-OHT preparation and delivery

We dissolved 10 mg of 4-OHT (Sigma, Cat# H6278-10MG) in 500 µL ethanol (100%) (20 mg/mL stock) first by vortexing and then sonicating. We then add autoclaved corn oil (1:4) to dissolve 4-OHT (previously heated to 45℃) to 5 mg/mL and sonicate until solution cloudiness clears. As a final step, vacuum centrifuge for 10 min to evaporate the alcohol from the final injection solution. Male FosTRAP (FosCreER+/-, FosCreER+/-, Ai9+/-) mice were used. Mice were single housed and gently handled for 7–10 days prior to the experiment to minimize the unwanted labeling of neurons associated with stress of handling. On the experiment day, mice were given 4-OHT 20 mg/kg in their homecage environment. 60 min post 4-OHT, we injected either saline or chloroquine (200 µg/50 µL) subcutaneously in the nape of the neck to TRAP neurons that are activated by pruritic stimuli. In FosCreER+/-, Ai9 ±mice, robust tdTomato expression was seen 1 week post TRAPing. In the FosCreER ± mice injected with the optogenetic or chemogenetic constructs, robust labeling was seen 4 weeks post TRAPing. All the TRAPs for behavioral experiments were performed between October and March, between 9.00 am and 1.00 pm.

## Pruritic agent-induced scratching behaviors

As previously described by our group (*O'Brien et al., 2013*; *Valtcheva et al., 2015*), the nape of the neck of mice was shaved 1 day prior to experiments. Mice were then placed in clear plexiglass

behavioral boxes for at least 2 hr for acclimation. For chemogenetic manipulations, CNO was administered before placing the mice in the plexiglass behavioral boxes and chloroquine (200 µg/50 µL, nape of the neck)-induced scratching behavior was performed 90 min after the CNO administration.

## Pain behavior assessment

Mechanical sensitivity was measured by counting the number of withdrawal responses to 10 applications of von Frey filaments (North Coast Medical, Inc, Gilroy, CA; 0.02, 0.08, 0.32 and 1.28 g von Frey filaments) to both hindpaws as described (*Samineni et al., 2017b*). Each mouse was allowed at least 15 s between each application and at least 5 min between each size filament. Animals were acclimated to individual boxes on a plastic screen mesh for at least 1 hr before testing. The Hargreaves test was performed to evaluate heat sensitivity thresholds as previously described (*Samineni et al., 2017a*). Briefly, we measured latency of withdrawal to a radiant heat source (IITC Life Science, Model 390). We applied the radiant heat source to both hindpaws and measured the latency to evoke a withdrawal. Three replicates were acquired per hindpaw per mouse and values for both paws were averaged.

## Open-field test

Before testing, mice were habituated to the test room in their home cages for 2 hr. Control and mice injected with either hM3Dq, hM4Di or ChR2 in the CeA were then placed in the open field during individual trials and allowed to freely explore after the experimenter exited the room (behaviors were video recorded). Open field locomotor activity was assessed in a square enclosure (55 × 55 cm) within a sound attenuated room for 30 min (*Shin et al., 2017*). Total distance traveled and movements were video recorded and analyzed using Ethovision XT (Noldus Information Technologies, Leesburg, VA).

## Elevated zero maze

Anxiety was measured in low-light conditions (~20 lux) using a modified zero maze (Stoelting Co., Wood Dale, IL) placed 70 cm off of the ground and consisting of two closed sections (wall height, 30 cm) and two open sections (wall height, 1.3 cm) on a circular track (diameter of track, 60 cm) (*Montana et al., 2011*). On the experiment day, mice were habituated to testing room for 1 hr before beginning of the behavioral session. For hM3Dq- and hM4Di-injected mice 60 min after CNO injection, mice were placed individually at the intersection of the closed/open area of the zero maze for a 6 min trial. For Chr2 and eYFP FosTRAP mice, mice were connected to the fiber optic and placed at the intersection of the closed/open area of the zero maze for a 6 min trial. Mice received 20 Hz (5 ms width) photostimulation for the duration of the EZM trial. Movement during the trial was video recorded using digital camera (Floureon HD) mounted on the ceiling of the room. Total distance traveled, number of entries into open sections and time spent in the open sections were scored, video recorded and analyzed using Ethovision XT (Noldus Information Technologies).

## Real-time place aversion testing

Place aversion was tested in a custom-designed two-compartment chamber (52.5 × 25.5 × 25.5 cm) with a layer of corn cob bedding (*Shin et al., 2017*). Each mouse was placed in the neutral area of the chamber and given free access to roam across both chambers. Activity was continuously recorded through a video camera for a period of 20 min. Entry into light-paired chamber triggered constant photostimulation at either 5 Hz, 10 Hz, 20 Hz or 30 Hz (473 nm, 5 ms pulse width, ~10 mW light power). Entry into the other chamber terminated the photostimulation. Photostimulation was counterbalanced across mice. 'Time-in-chamber' and heatmaps were generated for data analysis using Ethovision XT software (Noldus Information Technology).

## Conditioned place aversion

CPA was performed using an unbiased, counterbalanced three-compartment conditioning apparatus as described (*Land et al., 2009*). Each chamber had a unique combination of visual properties (one side had black and white vertical walls, whereas the other side had black and white horizontally striped walls). On the pre-conditioning day (day 1), mice were allowed free access to all three chambers for 20 min. Behavioral activity in each compartment was monitored and recorded with a video

camera and analyzed using Ethovision 8.5 (Noldus Information Technology) or ANY-Maze software. Mice were randomly assigned to saline and chloroquine compartments and received a saline injection (50 μL) in the nape of the neck and on the mouse caudal back, in the morning and a chloroquine injection (200 μg/50 μL) in the nape of the neck and on the mouse caudal back in the afternoon, at least 4 hr after the morning training on three consecutive days (days 2–4). To enhance the association of chloroquine-induced scratching behavior with the paired chamber, we administered chloroquine and left the mice in their holding cage for 4 min, then placed them in the paired chamber during the time of the peak scratching response (20 min in the chamber). To assess for place aversion, the mice were then allowed free access to all three compartments on day 5 for 30 min (*Tzschentke, 2007*). Scores were calculated by subtracting the time spent in the chloroquine-paired compartment, post-test minus the pre-test. To test the effect of DREADD hM4Di activation on chloroquine-induced place aversion, mice injected with AAV5-DIO-hM4Di–mCherry and AAV5-DIO-mCherry were allowed free access to all three chambers for 30 min on the pre-conditioning day (day 1). On days 2–4, both cohorts received a saline injection (50 μL) in the nape of the neck and on the mouse caudal back, and this chamber was paired with systemic saline injection 1 hr before they were placed in the compartment in the morning and a chloroquine injection (200 μg/50 μL) in the nape of the neck and on the caudal back, and this chamber was paired with systemic CNO injection 1 hr before they were placed in the compartment in the afternoon. To test the effect of DREADD hM4Di activation on chloroquine-induced place aversion, the mice were allowed free access to the three compartments on day 5 for 30 min. Scores were calculated by subtracting the time spent in the chloroquine-paired compartment, post-test minus the pre-test.

## Operant conditioning

Mice are food-deprived to reach 90% of their body weight and trained to nose poke for sucrose pellets for 7 days during daily 60 min sessions in a modular test chamber (Med Associates) on a fixed ratio 1 (FR1) schedule of reinforcement as previously described by *Seo et al., 2016*, *Shin et al., 2017*. A correct nose poke response in the active hole resulted in a sucrose pellet delivery where an incorrect nose poke within the inactive hole resulted in no sucrose pellet. On the experiment day, mice were administered CNO followed by a 60 min operant self-stimulation session. To determine if DREADD manipulation of FosTRAPed CeA neurons has any effect on FR1 schedule of reinforcement, mice were given free access to nose poke the ports, three successive nose pokes (FR3) to the active portal rewarded the mouse a sucrose pellet delivery where an incorrect nose poke within the inactive hole resulted in no sucrose pellet. On the experiment day, mice were administered CNO followed by a 60 min operant self-stimulation session to determine if DREADD manipulation of FosTRAPed CeA neurons has any effect on fixed ratio 3 (FR3) schedule of reinforcement.

## Feeding behavior

Mice were given free access to a novel empty cage prior to the experiment day. Mice were food-deprived overnight prior to the experiment day (*Cai et al., 2014*). Mice were reintroduced into the same empty cage they had access to the prior day but with food pellets and allowed to feed freely for 20 min on the experiment day. At the end of the session, weight of the food pellet and the food debris left on the cage floor was measured to calculate the food intake. To determine whether FosTRAPed CeA neurons modulate feeding behaviors, mice were injected with CNO 60 min before the feeding test. Feeding tests were performed between 2 pm and 7 pm.

## Fiber photometry

For in vivo calcium imaging of CeA GABAergic neurons, we injected the CeA of Vgat-Cre mice with Cre-dependent GCaMP6s (AAV-DJ EF1a-DIO-GCaMP6s, $3 \times 10^{13}$ particles/mL, Stanford vector core). Fiber optic probes were unilaterally implanted above the right CeA ($-1.24$ mm from bregma, $\pm 2.84$ mm lateral from midline and 4.4 mm ventral to skull). After 4 weeks of viral expression, mice were handled and acclimated by tethering as will occur during imaging sessions for 7 days in the test behavioral chamber. On the test day, mice were habituated with the tethered fiber optic patch cord (0.48 NA, BFH48-400, Doric Lenses) in the test chamber (15 × 15 cm) for 60 min and then injected with chloroquine (200 μg/50 μL) in the nape of the neck and recordings were performed.

A fiber optic patch cord was used to connect to the fiber implant and deliver light to excite and record the GCAMP signal using a custom-built fiber photometry rig, built with some modifications to previously described specifications (*Cui et al., 2013*). Fluorescence excitation was provided by two LEDs at 211 and 537 Hz to avoid picking up room lighting (M405FP1, M470F1; LED driver: LEDD1B; Thorlabs). Light was bandpass filtered (FMC1 + (405/10) -(475/28)_(525/45)_FC, Doric Lenses) and delivered to the CeA to excite GCaMP6s. The emitted light was bandpass filtered (FMC1 +_(405/10) - (475/28)_(525/45)_FC, Doric lenses) and sent to a photoreceiver to detect the signal (Newport, 2151). The signal from the photoreceiver was recorded using a RZ5P real-time processor (TDT). Data were acquired at 10 kHz and then demodulated at 211 and 537 Hz. The demodulated signal was then low-pass filtered (4 Hz) in a custom MATLAB script. The extracted 405 nm signal was then scaled to fit the GCaMP signal for the recording session. To isolate the movement-corrected GCaMP signal from channel, we subtracted the signal at 405 nm from the 475 nm GCaMP signal. dF/F was obtained by dividing the final signal with its mean value. Behavioral event time-stamps associated with chloroquine-evoked scratching behavior were scored and aligned with GCaMP signal in the MATLAB script to create pre- and peri-stimulus time bins. To obtain pre- and peri-stimulus chloroquine-evoked scratching events, if the scratching events happened close to each other (in a 30 s window), they were combined and scored as one bout. Z-score was obtained by sub-tracting the mean of the GCaMP signal from the bin value of the GCaMP signal and dividing it with the standard deviation of the bin value of the GCaMP signal.

## Acute slice electrophysiology

To determine the functional effects of chemogenetic manipulations in the chloroquine FosTRAPed CeA neurons, we performed targeted whole-cell patch-clamp recordings in acute coronal slices from cFos-Cre mice expressing either hM3Dq or hM4Di receptors as previously described (*Samineni et al., 2017a*). Mice used for electrophysiology and behavioral studies were between 8 and 16 weeks of age. Three weeks after viral injections, we performed chloroquine TRAP and waited 3 weeks for expression of hM3Dq or hM4Di in the CeA. Coronal slices containing the CeA were prepared and CeA neurons were visualized through a 40× objective using IR-DIC micros-copy on an Olympus BX51 microscope, and mCherry+ cells were identified using epifluorescent illu-mination with a green LED (530 nm; Thorlabs), coupled to the back-fluorescent port of the microscope. Whole-cell recordings of itch FosTRAPed CeA neurons expressing hM3Dq-mCherry and hM4Di-mCherry were performed using a Heka EPC 10 amplifier (Heka) with Patchmaster software (Heka). Following stable 5 min whole-cell recordings (baseline), the effects of either hM3Dq or hM4Di receptor activation on cellular excitability were isolated by blocking AMPA/KARs (10 μM NBQX, Abcam), NMDARs (50 μM D-APV, Abcam), GABAARs (100 μM picrotoxin, Abcam), and GABABRs (50 μM baclofen, Abcam), and aCSF solution containing 10 μM CNO added to the antag-onist cocktail above was bath applied to the brain slice.

## Immunohistochemistry

Adult mice were deeply anesthetized using a ketamine/xylazine cocktail and then perfused with 20 mL of phosphate-buffered saline (PBS) and 4% paraformaldehyde (weight/volume) in PBS (PFA; 4°C). *For Fos staining:* To determine the causal contribution of the CeA neuron in itch processing, we gave chloroquine to the nape of the neck and 90 min later mice were perfused. To verify whether chloroquine TRAPed tdTomato+ CeA neurons are faithfully TRAPed to pruritic stimulus and rule out non-specific labeling, 1 week after the TRAP, we gave a second chloroquine injection, and 90 min later mice were perfused. Brains were carefully removed, post fixed in 4% PFA overnight and later cryoprotected by immersion in 30% sucrose for at least 48 hr. Tissues were mounted in OCT while allowing solidification of the mounting medium at −80°C. Using a cryostat, 30 μm tissue sections were collected and stored in PBS1 × 0.4% sodium azide at 4°C. After washing the sections in PBS1×, we blocked using 5% normal goat serum and 0.2% Triton-X PBS 1× for 1 hr at room temper-ature. Primary antibodies against mCherry (Rabbit, Clontech, 632543; 1/1000), GFP (Chicken Mono-clonal anti-GFP, Aves A11122; 1/2000) and cFos (Rabbit monoclonal anti-phospho-cFos, Cell Signaling Ser32 D82C12; 1:2000) were diluted in blocking solution and incubated overnight at 4°C. After three 10 min washes in PBS1×, tissues were incubated for 1 hr at room temperature with sec-ondary antibodies (Life Technologies: Alexa Fluor488 donkey anti rabbit IgG [1/500]; Alexa Fluor

488 goat anti rabbit [1/500]; Alexa Fluor 555 goat anti mouse [1/500]; Alexa Fluor 555 goat anti rabbit [1/500]) and Neurotrace (435/455 nm, 1/500) at room temperature. Three PBS1× washes followed before sections were mounted with Vectashield (H-1400) hard-mounting media and imaged after slides cured. Images were obtained on a Nikon Eclipse 80i epifluorescence microscope.

## Tissue preparation for RNA-seq and Fac sorting

Animals (8–10-week-old, 7–10 days post TRAP) were used for this experiment to ensure robust Ai9 reporter expression, while assuring fully developed brains. RNA-seq of the TRAPed neurons was performed using protocols modified from prior published work to improve neuronal survival (*Arttamangkul et al., 2006*; *Guez-Barber et al., 2012*; *Hempel et al., 2007*). Animals were anesthetized with ketamine cocktail, perfused with aCSF (124 mM NaCl, 24 mM NaHCO$_3$, 12.5 mM glucose, 2.5 mM KCl, 1.25 mM NaH$_2$PO$_4$, 2 mM CaCl$_2$, 1 mM MgCl$_2$, 5 mM HEPES, pH 7.4, 300–310 mOsm) and decapitated for brain removal. The brains were allowed to rest in cold oxygenated (95% O$_2$/5% CO$_2$) aCSF and then sliced coronally using a vibratome (Leica VT1000 S). Brain slices (400-µm-thick sections) were collected and kept in cold oxygenated aCSF. Tissues were micro-dissected under a microscope (Leica S9i) using a reusable 0.5 mm biopsy punch (WPI 504528). HBSS+H and Papain solution (45U, Worthington, Lakewood, NJ) was incubated for 5 min at 37˚, followed by the addition of tissue punches for 10–15 min. Tissue punches were then transferred to ice, and mechanical trituration of tissue punches was performed using ~600, 300 and 150 µm fire-polished Pasteur pipettes. The resulting cell suspension was then centrifuged at 5k RPM for 5 min to obtain a pellet, and cells were resuspended in fresh aCSF. This process happened twice to wash any remnants of Papain. Cells were ultimately resuspended for FACS sorting into cold oxygenated aCSF and kept on ice for the duration of the experiment.

Cell suspensions were kept cold throughout the FACS, and cells were sorted in aCSF. In order to determine gating criteria for selecting cell bodies while excluding debris, we performed FACS on fixed/permeabilized neurons stained with Neurotrace 435/455 nm (Nissl stain). Samples were treated with 2% PFA for 20 min, pelleted down for 5 min at 5k RPM, and then resuspended in PBS1 × 0.3% Triton X-100. This processed was done an additional time to get rid of any remnant of PFA. Cells were then resuspended in aCSF and incubated with Neurotrace 435/455 (Thermo Fisher, #N211479) for FACS sorting. We gated for events that had high levels of Neurotrace, and then mapped these events in the scatter plot (forward scatter [FSC] vs. side scatter [SSC]). We were able to map events that had high Neurotrace expression to a small subset of events, which represent the population of cell bodies and not debris. In addition, this population was sensitive to PFA fixation and labeling with the nuclear staining DAPI or 7-AAD, which is characteristic of post-fixative dead cells. As for DAPI/7-AAD (dead) control samples, these were incubated in 2% PFA for 20 min, pelleted down for 5 min at 5k RPM, and then resuspended in PBS1 × 0.3% Triton X-100; this process was done an additional time to get rid of any remnant PFA. Cells were then resuspended in aCSF and incubated with DAPI (1:1000 dilution of 1 mg/mL DAPI, Thermo Fisher, #62248 or 7-AAD 7-Aminoactinomycin D, A1310, Thermo Fisher) for FACS sorting. We performed control experiments to set the appropriate gates for florescence, Ai9 (tdtomato) expression. Negative control samples were obtained from c57BL6/J animals, while positive controls were obtained from Vgat Ai9. FosCre × Ai9 brains were used for isolation of the neuronal population of interest. The CeA was dissociated as previously described (*Guez-Barber et al., 2012*), and cells were sorted into a 96-well plate. Up to a maximum of 50 cells were sorted into one well filled with 9 µL of Clontech lysis buffer (Single-cell lysis buffer 10×, #635013 Takara Bio) + 5% RNAse inhibitor (40 U/µL, Promega RNAsin inhibitor N2511). Samples were then transferred to a tube for processing by our Genome Technology Access Center (GTAC) core facility. ds-cDNA was prepared using the SMARTer Ultra Low RNA kit for Illumina Sequencing (Takara-Clontech) per manufacturer's protocol using the lysis buffer as substrate for the reaction. cDNA was fragmented using a Covaris E220 sonicator using peak incident power 18, duty factor 20%, cycles/burst 50, time 120 s. cDNA was blunt ended, had an A base added to the 3′ ends and then had Illumina sequencing adapters ligated to the ends. Ligated fragments were then amplified for 15 cycles using primers incorporating unique index tags. Fragments were sequenced on an Illumina HiSeq-3000 using single reads extending 50 bases.

## RNA-seq

RNA-seq reads were aligned to the Ensembl top-level assembly with STAR version 2.0.4b. Gene counts were derived from the number of uniquely aligned unambiguous reads by Subread:feature-Count version 1.4.5. Transcript counts were produced by Sailfish version 0.6.3. Sequencing performance was assessed for the total number of aligned reads, total number of uniquely aligned reads, genes and transcripts detected, ribosomal fraction known junction saturation and read distribution over known gene models with RSeQC version 2.3.

All gene-level and transcript counts were then imported into the R/Bioconductor package EdgeR, and TMM normalization size factors were calculated to adjust for samples for differences in library size. Ribosomal features as well as any feature not expressed in at least the smallest condition size minus one sample were excluded from further analysis, and TMM size factors were recalculated to created effective TMM size factors. The TMM size factors and the matrix of counts were then imported into R/Bioconductor package Limma, and weighted likelihoods based on the observed mean-variance relationship of every gene/transcript and sample were then calculated for all samples with the voom with quality weights function. Generalized linear models were then created to test for gene-/transcript-level differential expression. Differentially expressed genes and transcripts were then filtered for p-values less ≤0.001.

The biological interpretation of the genes found in the Limma results was then queried for global transcriptomic changes in known Gene Ontology (GO) and KEGG terms with the R/Bioconductor packages GAGE and Pathview. Briefly, GAGE measures for perturbations in GO or KEGG terms based on changes in the observed log2-fold changes for the genes within that term versus the background log2-fold changes observed across features not contained in the respective term as reported by Limma. For GO terms with an adjusted statistical significance of FDR $\leq$ 0.05, heatmaps were automatically generated for each respective term to show how genes co-vary or co-express across the term in relation to a given biological process or molecular function. In the case of KEGG curated signaling and metabolism pathways, Pathview was used to generate annotated pathway maps of any perturbed pathway with an unadjusted statistical significance of p-value≤0.05.

To find the most critical genes, the raw counts were variance stabilized with the R/Bioconductor package DESeq2 and were then analyzed via WGCNA with the R/Bioconductor package WGCNA. Briefly, all genes were correlated across each other by Pearson correlations and clustered by expression similarity into unsigned modules using a power threshold empirically determined from the data. An eigengene was then created for each de novo cluster, and its expression profile was then correlated across all coefficients of the model matrix. Because these clusters of genes were created by expression profile rather than known functional similarity, the clustered modules were given the names of random colors where gray is the only module that has any preexisting definition of containing genes that do not cluster well with others. The information for all clustered genes for each module was then combined with their respective statistical significance results from Limma to determine whether or not those features were also found to be significantly differentially expressed. Raw and analyzed data can be found at GEO: GSE130268.

## Fluorescence in situ hybridization

C57BL/6J mice were injected with chloroquine on the nape of the neck. Thirty minutes post-chloroquine administration, mice were rapidly decapitated, brains were dissected and flash frozen in −50°C 2-methylbutane and stored at −80°C for further processing (*Samineni et al., 2017a*). Coronal sections of the brain corresponding to the CeA were cut at 15 µM at −20°C and thaw-mounted onto Super Frost Plus slides (Fisher). Slides were stored at −80°C until further processing. FISH was performed according to the RNAScope 2.0 Fluorescent Multiple Kit v2 User Manual for Fresh Frozen Tissue (Advanced Cell Diagnostics, Inc). Slides containing CeA sections were fixed in 4% PFA, dehydrated and pretreated with protease IV solution for 30 min. Sections were then incubated with target probes for mouse cFos (mm-Fos, catalog number 316921, Advanced Cell Diagnostics), Ntsr2 (mm-Ntsr2, catalog number 452311, Advanced Cell Diagnostics), GPR88 (mm-GPR88, catalog number 317451, Advanced Cell Diagnostics), Penk (mm-Penk, catalog number 318761, Advanced Cell Diagnostics), Gabarg1 (mm-Gabarg1, catalog number 501401, Advanced Cell Diagnostics), Oprm1 (mm-Oprm1, catalog number 315841, Advanced Cell Diagnostics) and Chrm1 (mm-Chrm1, catalog number 495291, Advanced Cell Diagnostics). Following probe hybridization, sections underwent a

series of probe signal amplification steps (AMP1–4) followed by incubation of fluorescent probes (Opal 470, Opal 570, Opal 670), designed to target the specified channel associated with the probes. Slides were counterstained with DAPI and coverslips were mounted with Vectashield Hard Set mounting medium (Vector Laboratories). Images were obtained on a Leica TCS SPE confocal microscope (Leica), and Application Suite Advanced Fluorescence (LAS AF) software was used for analyses. To quantify number of cFos+ve cells, we counted DAPI-stained nuclei that coexpress minimum of five cFos puncta as a cFos+ve cell. We did not include any cFos puncta that does not overlay on top of the DAPI-stained nuclei as part of our analysis.

## Statistics

Throughout the study, researchers were blinded to all experimental conditions. Exclusion criteria for our study consisted of a failure to localize expression in our experimental models or off-site administration of virus or drug. At least three replicates measurements were performed and averaged in all behavioral assays. The number of animals used is indicated by the 'N' in each experiment. When paired t test was used for comparing paired observations, we evaluated for normality using the D'Agostino and Pearson omnibus normality test for all datasets. Therefore, only when normality could be assumed we used a parametric test to analyze out data. If normality could not be assumed, a nonparametric test or a Wilcoxon matched pairs text was used to evaluate differences between the means of our experimental groups. Two-way ANOVA was used for comparing between different control and treatment groups. Bonferroni's *post hoc* tests were used (when significant main effects were found) to compare effects of variables. A value of $p < 0.05$ was considered statistically significant for all statistical comparisons.

## Acknowledgements

This work was funded by NINDS R01NS106953 and R01DK116178 to RWG, the Urology Care Foundation Research Scholars Program and Kailash Kedia Research Scholar Award and NIDDK Career development award (K01 DK115634) to VKS, and the Medical Scientist Training Program (MSTP) Grant T32GM07200 and NINDS NRSA 5F31NS103472-02 to JGGR. We Kenneth M Murphy and his lab for assistance with FACS. We thank Sherri Vogt for her assistance with mouse colony maintenance and genotyping. We would like to thank Dr. Jordan G McCall for helpful discussion with the manuscript and experimental help with slice electrophysiology. We also thank Daniel Castro and Adrian Gomez for their help with reward-seeking experiments. We would like to thank all the Gereau lab members for their help with manuscript preparation.

## Additional information

### Funding

| Funder | Grant reference number | Author |
|---|---|---|
| National Institute of Neurological Disorders and Stroke | R01NS106953 | Robert W Gereau |
| National Institute of Diabetes and Digestive and Kidney Diseases | R01DK116178 | Robert W Gereau IV |
| National Institute of Diabetes and Digestive and Kidney Diseases | K01 DK115634 | Vijay K Samineni |
| National Institute of Neurological Disorders and Stroke | 5F31NS103472-02 | Jose G Grajales-Reyes |
| National Institute of Diabetes and Digestive and Kidney Diseases | R01 DK128475 | Vijay K Samineni |

The funders had no role in study design, data collection and interpretation, or the decision to submit the work for publication.

## Author contributions
Vijay K Samineni, Conceptualization, Data curation, Formal analysis, Supervision, Funding acquisition, Validation, Investigation, Visualization, Methodology, Writing - original draft, Project administration, Writing - review and editing; Jose G Grajales-Reyes, Funding acquisition, Validation, Investigation, Writing - review and editing; Gary E Grajales-Reyes, Formal analysis, Investigation; Eric Tycksen, Formal analysis, Investigation, Visualization; Bryan A Copits, Data curation, Formal analysis, Investigation, Visualization; Christian Pedersen, Software, Formal analysis; Edem S Ankudey, Julian N Sackey, Sienna B Sewell, Investigation; Michael R Bruchas, Resources, Software, Supervision; Robert W Gereau, Conceptualization, Resources, Data curation, Supervision, Funding acquisition, Project administration, Writing - review and editing

## Author ORCIDs
Vijay K Samineni (iD) https://orcid.org/0000-0002-9491-2793
Eric Tycksen (iD) https://orcid.org/0000-0001-6362-0141
Bryan A Copits (iD) https://orcid.org/0000-0003-3732-890X
Michael R Bruchas (iD) http://orcid.org/0000-0003-4713-7816
Robert W Gereau (iD) https://orcid.org/0000-0002-5428-4251

## Ethics
Animal experimentation: This study was performed in strict accordance with the recommendations in the Guide for the Care and Use of Laboratory Animals of the National Institutes of Health. All of the animals were handled according to approved institutional animal care and use committee (IACUC) of Washington University School of Medicine (approved protocol 20-0078).

## Decision letter and Author response
Decision letter https://doi.org/10.7554/eLife.68130.sa1
Author response https://doi.org/10.7554/eLife.68130.sa2

# Additional files

## Supplementary files
• Transparent reporting form

## Data availability
Sequencing data have been deposited in GEO under accession codes GSE130268.

The following dataset was generated:

| Author(s) | Year | Dataset title | Dataset URL | Database and Identifier |
|---|---|---|---|---|
| Samineni VK | 2019 | Transcriptional Idenetity of Itch-activated Central Amygdala Neurons | https://www.ncbi.nlm.nih.gov/geo/query/acc.cgi?acc=GSE130268 | NCBI Gene Expression Omnibus, GSE130268 |

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
