## [Decision Letter]

**Acceptance summary:**

We consider that this work will be of general interest to neuroscientists, especially those studying how the brain processes itch stimuli and controls itch-related behavior. Here, the authors employed an elegant and multidisciplinary approach to conclude that specific cells in the central nucleus of the amygdala (and their communication with other parts of the brain) play an important role in itching (pruritic) behavior. Importantly, the data presented in the manuscript strongly support the authors' conclusions.

**Decision letter after peer review:**

Thank you for submitting your article "Cellular, Circuit and Transcriptional Framework for Modulation of Itch in the Central Amygdala" for consideration by *eLife*. Your article has been reviewed by 3 peer reviewers, and the evaluation has been overseen by a Reviewing Editor and Laura Colgin as the Senior Editor. The following individuals involved in review of your submission have agreed to reveal their identity: Robert Sears (Reviewer #1); Mark A Hoon (Reviewer #2).

Essential Revisions:

The reviewers consider this study to be a strong candidate for publication in *eLife*. However, they have made several recommendations to improve the overall quality of the manuscript. These are summarized below. Notice that while the reviewers raised important questions, particularly regarding the differential contribution of CeA neurons to itch sensation and its associated behaviors (e.g. scratching), it is our collective opinion that these and the rest of the critiques can (and must) be addressed in writing or through modifications to the draft itself. No further experimental work is required.

*Reviewer #1 (Recommendations for the authors):*

1. It is not clear what the authors mean by 'active avoidance' and how it relates to itch-related behaviors or "strong urges to scratch".

2. Elevated zero maze is not active avoidance. It is not a learned adaptive behavior. It is innate and closer inhibitory avoidance. Therefore, these studies do not show "the necessity and sufficiency of itch-activated neurons…… in active avoidance".

3. In the Abstract it says "……viral tracing experiments demonstrate that these neurons send critical projections to the periaqueductal gray to mediate modulation of itch". This does not encapsulate their circuit function studies. I suggest the authors further specify that optogenetic manipulations demonstrate that these projections are 'critical'.

4. It is stated in the results that freezing and flight were not affected by the manipulations, but no freezing or flight data is presented. Total distance traveled is shown in one case (Figure 3) and velocity is measured (Supplementary Figure 6), but these measures say nothing about defensive behaviors. EZM and OFT aren't the proper paradigms to assess these defensive responses, which CeA (or at least some CeA microcircuits and projects) is well described to be involved in.

5. The authors state that ChR2 stimulation of TRAPed neurons led to spontaneous full body scratching and grooming bouts (page 6., paragraph 2). Are the animals scratching (and/or grooming) spontaneously and excessively in the closed arms of the EZM, around the walls of the OFT, etc. during optogenetic and DREADD stimulation?

6. The authors need to address why all studies are done in male animals.

7. A saline-injected control group is used in the initial TRAP experiment and presented in Supplementary Figure 2. This is an important control considering that a painful needle stick may elicit c-Fos activity in the pain-sensitive CeA. In support of this possibility, Supplementary Figure 2 shows a substantial number of TRAPed cells in the saline control group (about 20% of what is shown for the chloroquine-injected group). Also, results of the von Frey test (Supplementary Figure 5c) show that hM3D activation of chloroquine-TRAPed cells may increase pain sensitivity. However, this control is missing in other experiments. Most notably, there is no saline control in the RNASeq experiments. This seems important to say anything about itch-responsive-specific cellular enrichment. Thus, the authors should acknowledge that their TRAP and RNASeq results may include pain-sensitive CeA neurons.

8. Saying an animal has an itch is tricky (as does saying an animal is experiencing 'fear'). What can be said is that exhibit scratching behavior. This is not my specific field, but it would seem to warrant further discussion.

9. I highly recommend removing the term 'active avoidance' because it is not active avoidance you are testing. I do understand, though, that active avoidance is an emerging (and important) focus of the pain literature.

10. Given the availability of TRAP, it is surprising that itch-TRAPed neurons are not observed in photometry experiments. Was there a reason why this was not done?

11. I think that it is very difficult to separate the pain component of these findings from the 'itch' component. I suggest that this be acknowledged.

12. The paper by Sanders, K.M. et al. (J. Neuroscience, 2019) seems highly relevant. I don't see it discussed or referenced.

*Reviewer #2 (Recommendations for the authors):*

The changes I would recommend can be addressed by alterations to the existing text.

• Why were CeA VGat-inhibitory neurons examined (figure 1)? The Results give no logical or practical reason as to why these and not some other population of neurons were studied, and the study would have been better if TRAPPED neurons had been imaged. In the absence of this experiment (I am not sure this is needed since this experiment is not absolutely required to support the conclusions of the study), the limitation of likely examining a heterogeneous population of neurons all of which may not be involved in itch, should be acknowledged. Further, there should be some discussion of the high variability of the reported increases in calcium (figure 1e). In addition, these calcium responses do not distinguish between the itch responses and the counter-stimulation nociception triggered by the scratch. Lastly, this figure might more logically be placed at the end of the Results instead of being the first figure.

• The manuscript does not mention whether there are other projections of itch TRAPPED CeA-neurons. Other projections should be stated and the potential for these other neuronal projection also contributing to scratching should be acknowledged. Indeed, these additional projections may contribute to a "itch code" and could thereby reconcile the fact that there are many CeA-PAG projections involved in driving multiple different aversive responses (not just itch).

*Reviewer #3 (Recommendations for the authors):*

In its completeness, the manuscript raises an important number of open questions in the field, and I would like to encourage the authors to identify these more clearly in their discussion, as they could set out a pathway along which this field may develop further, notably:

1. If the activation of the neurons that are activated by itch causes non-directed scratching behavior, the question arises what is needed for DIRECTED scratching behavior to occur? One could imagine that a further activation of (higher order) circuitries is involved notably:

a. A topological representation of the body surface to localize the itch.

b. A motor system that topologically directs the scratching activity to the localization of the itch.

Which areas should therefore be involved in the complete set of behavior that they observed? And what precise role does the CeA play in this? How do these two systems work in parallel? In this context it might be interesting to consider the theoretical framework of Balleine and Killcross (TINS, 2006) who propose a parallel processing between CeA and basolateral amygdala (BLA) (for fear behavior). In their framework, the CeA circuitry provides the general motivational drive and the BLA (though its connections with the cortex) the more specific detection of sensory stimulus. Could the authors see a possibility to integrate this in their discussion?

2. The parallel already drawn with fear learning, the authors make a beautiful genetic characterization of the neurons that are identified. From fear learning field, a lot has become known about the internal circuitry within the CeA and the (partial) genetic identification of neurons involved: those that trigger the freezing behavior (the somatostatin expressing neurons) and those that inhibit the freezing behavior (PKCdelta neurons, Ciocchi et al., Haubensak et al., Nature 2010) and those that trigger escape (CRF neurons, Fadok et al. Nature, 2018). The authors do mention these different behaviors, but do not discuss the molecular markers that identify them (CRF, PKC δ+, somatostatin). Did the authors find any of the markers that have been used to characterize these neurons in their genetic analyses? How did these markers distribute across the neuronal populations that they identified? What about the neurons identified to be activated by isoflurane? Do they co-segregate with these itch-activated neurons'?

3. Finally, the most intriguing question poses itself what would be the role of the scratching and how would scratching affect the sensory sensation of the itching? Generally believed it seems that scratching on the spot of the itch, reduces the itching behavior. In that case would mice that are able to scratch , show a lower activation of the CeA neurons, compared to those that are prevented from scratching the itching spot? And furthermore, is this (negative) feedback loop only existing at the behavioral level outside of the body or is it possible that also at the circuitry level of the CeA a representation of this motor-sensory feedback loop exists (by various types of inhibitory connections that have already been described been other cell-types?). In this same context, it seems odd that neurons that are activated by the itching would be exactly the same neurons that are at the origin of the scratching behavior. An animal may choose not to scratch, for example when facing a dangerous situation where freezing would be of essential for survival (and one can recall a number of Hitchcock-genre videos where actors have faced very similar situations). So somewhere an active inhibition most take place and the CeA, with its many inhibitory neuronal connections between local cell populations might play an important role. However, this would imply different populations for itch and scratch possibly within the CeA. Could the authors discuss this within the context of what is already known in the field?

---

## [Author Response]

Essential Revisions:The reviewers consider this study to be a strong candidate for publication in eLife. However, they have made several recommendations to improve the overall quality of the manuscript. These are summarized below. Notice that while the reviewers raised important questions, particularly regarding the differential contribution of CeA neurons to itch sensation and its associated behaviors (e.g. scratching), it is our collective opinion that these and the rest of the critiques can (and must) be addressed in writing or through modifications to the draft itself. No further experimental work is required.Reviewer #1 (Recommendations for the authors):1. It is not clear what the authors mean by 'active avoidance' and how it relates to itch-related behaviors or "strong urges to scratch".

We do understand how our use of “active avoidance” can lead to confusion. Itch is an aversive sensory experience. In mice, pruritic stimuli (chloroquine and histamine) can produce robust place aversion (Mu and Sun, 2017 and Samineni et al., 2019). We interpreted this learned avoidance to pruritic stimuli as 'active avoidance'. As you pointed out, this can lead to confusion in interpreting our results. To mitigate any confusion, we have now removed any reference to active avoidance in the manuscript.

2. Elevated zero maze is not active avoidance. It is not a learned adaptive behavior. It is innate and closer inhibitory avoidance. Therefore, these studies do not show "the necessity and sufficiency of itch-activated neurons…… in active avoidance".

As mentioned above, we have now removed any reference to active avoidance in the manuscript.

3. In the Abstract it says "……viral tracing experiments demonstrate that these neurons send critical projections to the periaqueductal gray to mediate modulation of itch". This does not encapsulate their circuit function studies. I suggest the authors further specify that optogenetic manipulations demonstrate that these projections are 'critical'.

Thanks for this suggestion; we have now changed it to “Finally, viral tracing experiments demonstrate that these neurons send projections to the periaqueductal gray that are critical in modulation of itch.”

Our modifications to the manuscript: We have now changed abstract to refer to this, and the following text was added to page 2 line 10 to line 12 of the revised manuscript:

“Finally, viral tracing experiments demonstrate that these neurons send projections to the periaqueductal gray that are critical in modulation of itch.”

4. It is stated in the results that freezing and flight were not affected by the manipulations, but no freezing or flight data is presented. Total distance traveled is shown in one case (Figure 3) and velocity is measured (Supplementary Figure 6), but these measures say nothing about defensive behaviors. EZM and OFT aren't the proper paradigms to assess these defensive responses, which CeA (or at least some CeA microcircuits and projects) is well described to be involved in.

We agree that the distance and velocity travelled we have presented in our manuscript is not accurate method to quantify freezing and flight behaviors. We have used these data as a surrogate measure to measure freezing and flight behaviors. In the behavioral experiments we have conducted to assess itch and pain behaviors, we have not observed occurrence of freezing or flight behaviors. We have not formally attempted to quantify freezing or flight behaviors as we have failed to observe these behaviors.

Our modifications to the manuscript: We have now made changes in results in reference to this, and the following text was added to page 9 line 8 to page 9 line 11 of the revised manuscript:

“We used distance and velocity travelled as surrogate measures of freezing and flight behaviors. Although in our experiments assessing itch and pain behaviors we did not observe obvious freezing or flight behaviors, we did not more formally attempted quantify freezing or flight behaviors.”

5. The authors state that ChR2 stimulation of TRAPed neurons led to spontaneous full body scratching and grooming bouts (page 6., paragraph 2). Are the animals scratching (and/or grooming) spontaneously and excessively in the closed arms of the EZM, around the walls of the OFT, etc. during optogenetic and DREADD stimulation?

While we do see scratching behaviors in OFT, the resolution of the video we have obtained from our recordings is not accurate enough to quantify scratching behaviors from top view. In the EZM, ChR2 stimulated mice spent significant time in the closed arms, though we were able track them with Ethovision, it is not bright enough to visualize any scratching/grooming behaviors in the EZM. We are therefore unfortunately not able to comment specifically on these issues due to inability to quantify such behaviors in these recordings.

6. The authors need to address why all studies are done in male animals.

We acknowledge that sex is an important biological variable. In our initial study, we conducted a pilot experiment using small group of male and female mice, and we did not see any differences between both groups. As we have not seen any differences in our initial studies, we didn’t account for sex differences in our power analysis when we designed the comprehensive study. As this is a resource intense study we then proceeded to focus all our work on male mice.

Our modifications to the manuscript: We have now made changes in methods in reference to this, and the following text was added to page 23 line 7 to line 12 of the revised manuscript:

“We conducted a pilot experiment using both male and female mice. […] As this is a resource intensive study, we proceeded to focus the full study on a single sex, and in this case we used only male mice.”

7. A saline-injected control group is used in the initial TRAP experiment and presented in Supplementary Figure 2. This is an important control considering that a painful needle stick may elicit c-Fos activity in the pain-sensitive CeA. In support of this possibility, Supplementary Figure 2 shows a substantial number of TRAPed cells in the saline control group (about 20% of what is shown for the chloroquine-injected group). Also, results of the von Frey test (Supplementary Figure 5c) show that hM3D activation of chloroquine-TRAPed cells may increase pain sensitivity. However, this control is missing in other experiments. Most notably, there is no saline control in the RNASeq experiments. This seems important to say anything about itch-responsive-specific cellular enrichment. Thus, the authors should acknowledge that their TRAP and RNASeq results may include pain-sensitive CeA neurons.

This is an excellent point. We agree the possibility of saline TRAPped neurons and small proportion of chloroquine TRAPped neurons being pain responsive population that are activated by needle prick. We now have acknowledged this in our results and discussion.

Our modifications to the manuscript: We have now made changes in results in reference to this, and the following text was added to page 6 line 1 to 3 of the revised manuscript:

“This small population of saline TRAPped neurons could be due to the needle stick during the injection itself, and thus could label some pain responsive CeA neurons.”

8. Saying an animal has an itch is tricky (as does saying an animal is experiencing 'fear'). What can be said is that exhibit scratching behavior. This is not my specific field, but it would seem to warrant further discussion.

A very good point. It is difficult to interpret if the animal is having experiencing itch. Administration of known pruritic agents like chloroquine can lead to scratching in rodents. In our study, there are instances where CeA stimulation led to scratching behaviors, which we referred to as itch in the manuscript. To be more specific, we have removed any references to “mice are itching” or “itch TRAP” in the manuscript and now refer to these behaviors as scratching behaviors.

9. I highly recommend removing the term 'active avoidance' because it is not active avoidance you are testing. I do understand, though, that active avoidance is an emerging (and important) focus of the pain literature.10. Given the availability of TRAP, it is surprising that itch-TRAPed neurons are not observed in photometry experiments. Was there a reason why this was not done?

To understand causal relationship of CeA neurons in itch processing, we performed photometry recordings from the CeA Vgat neurons. As the majority of CeA neurons are GABAergic (Swanson and Petrovich, 1998), this approach will allow us to target just the CeA and avoid picking up photometry signals from neighboring BLA neurons, as could occur if we used non Cre-dependent GCaMP6. This experiment was done to determine whether CeA neurons were functionally active during pruritogen-evoked scratching. Seeing this activity, we then moved on to conduct the TRAP experiments to characterize the properties and function of pruritogen-activated neurons in the CeA. We agree that using TRAP mice for photometry could offer some additional insight into the activity of these CeA neurons in itch-scratch behaviors, the information gained would be unlikely to change the conclusions of the studies as they stand now.

Our modifications to the manuscript: We have now made changes in results in reference to this, and the following text was added to page 4 line 2 to 3 and Line 7 to 9 of the revised manuscript:

“As the majority of CeA neurons are GABAergic (Swanson and Petrovich, 1998), this approach allows us to target the CeA and avoid picking up photometry signals from neighboring BLA neurons, as could occur if we used non Cre-dependent GCaMP6.”

11. I think that it is very difficult to separate the pain component of these findings from the 'itch' component. I suggest that this be acknowledged.

We agree with your assessment. The current experimental design is not appropriate to differentiate itch processing completely form the pain component. We now have added this as part of our discussion.

Our modifications to the manuscript: We have now made changes in results in reference to this, and the following text was added to page 18 line 26 to page 19 line 2 of the revised manuscript:

“One other interesting observation from our sequencing dataset is the relative enrichment of Prkcd transcript in FosTRAP+ve vs FosTRAP-ve cells. Prkcd+ve cells in CeA have been shown to be involved in fear and pain processing. […] It is also possible that the needle stick associated with 4OHT injection could label a small population of CeA neurons involved in fear or pain processing, and this could impact our sequencing dataset to some extent.”

12. The paper by Sanders, K.M. et al. (J. Neuroscience, 2019) seems highly relevant. I don't see it discussed or referenced.

We now have cited this reference and incorporated into our discussion.

Reviewer #2 (Recommendations for the authors):The changes I would recommend can be addressed by alterations to the existing text.• Why were CeA VGat-inhibitory neurons examined (figure 1)? The Results give no logical or practical reason as to why these and not some other population of neurons were studied, and the study would have been better if TRAPPED neurons had been imaged. In the absence of this experiment (I am not sure this is needed since this experiment is not absolutely required to support the conclusions of the study), the limitation of likely examining a heterogeneous population of neurons all of which may not be involved in itch, should be acknowledged. Further, there should be some discussion of the high variability of the reported increases in calcium (figure 1e). In addition, these calcium responses do not distinguish between the itch responses and the counter-stimulation nociception triggered by the scratch. Lastly, this figure might more logically be placed at the end of the Results instead of being the first figure.

The rationale for imaging GABAergic neurons is described in response to reviewer 1 above.

The variability in Ca^2+^ signal that were detected by the photometry could be due to rapid nature of the itch-scratch cycle. To emphasize these differences we noticed, we have also included averaged GCaMP signal, which does not really capture all the variability in Ca^2+^ signal changes that we see when we plot them as individual trails in the heat map. Though the bulk signal detected from the photometry was able to tell us that the CeA is involved in meditating itch processing, photometry does not provide the resolution to distinguish between the itch and scratch responses. Future studies using in vivo single unit recordings or GRIN lens imaging could offer more opportunities in parsing our cellular level responses in itch-scratch behaviors.

• The manuscript does not mention whether there are other projections of itch TRAPPED CeA-neurons. Other projections should be stated and the potential for these other neuronal projection also contributing to scratching should be acknowledged. Indeed, these additional projections may contribute to a "itch code" and could thereby reconcile the fact that there are many CeA-PAG projections involved in driving multiple different aversive responses (not just itch).

In addition to projections to vPAG, we also see some projections in the BNST, lateral hypothalamus and faint projection in substantia nigra and PBN. We now mention this as part of our Results section and acknowledge this in our discussion.

Our modifications to the manuscript: We have now made changes in results in reference to this, and the following text was added to page 12 line 4 to line 6 of the revised manuscript:

“We also observed projections to the bed nucleus of stria terminalis (BNST), lateral hypothalamus and faint projection in substantia nigra and parabrachial nucleus (PBN).”

The following text was added to page 18 line 3 to line 6 of the revised manuscript:

“We also observed projections to the bed nucleus of stria terminalis (BNST), lateral hypothalamus and faint projection in substantia nigra and parabrachial nucleus (PBN) in addition to the vPAG projections. It is also possible these downstream regions could also play a critical role in different aspects of pruritis.”

Reviewer #3 (Recommendations for the authors):In its completeness, the manuscript raises an important number of open questions in the field, and I would like to encourage the authors to identify these more clearly in their discussion, as they could set out a pathway along which this field may develop further, notably:1. If the activation of the neurons that are activated by itch causes non-directed scratching behavior, the question arises what is needed for DIRECTED scratching behavior to occur? One could imagine that a further activation of (higher order) circuitries is involved notably:a. A topological representation of the body surface to localize the itch.b. A motor system that topologically directs the scratching activity to the localization of the itch.Which areas should therefore be involved in the complete set of behavior that they observed? And what precise role does the CeA play in this? How do these two systems work in parallel? In this context it might be interesting to consider the theoretical framework of Balleine and Killcross (TINS, 2006) who propose a parallel processing between CeA and basolateral amygdala (BLA) (for fear behavior). In their framework, the CeA circuitry provides the general motivational drive and the BLA (though its connections with the cortex) the more specific detection of sensory stimulus. Could the authors see a possibility to integrate this in their discussion?

Directed behavior related to sensory information could be organized at the level of sensory and motor cortex. Cortical areas have extensive direct descending projections to the dorsal and ventral horn of the spinal cord. These projections could orchestrate directed scratching behaviors. It is also possible that the CeA is not part of the neural pathways that translate into directed scratching behavior. Neurons in the CeA could receive itch related information from cortical inputs, which could be part of the neural pathways that mediate affective aspects of like motivation to scratch an itch or suppress itch to evade any immediate potential threat. Our data show that the CeA sends dense projections to the PAG, which in turn modulate spinal pruritic processing via RVM projections based on our prior work. There is still more to learn about how this information is organized. Recent work from Gao et 2019, Neuron, suggest that activating Tac1 neurons can drive robust scratching behaviors, this suggest that there might be parallel circuits downstream of the CeA than can evoke and inhibit itch evoked scratching.

Our modifications to the manuscript: We have now made changes in results in reference to this, and the following text was added to page 17 line 7 to line 24 of the revised manuscript:

“Activating CeA FosTRAP neurons resulted in spontaneous scratching and grooming behaviors directed all over the body, and thus were not restricted to the nape of the neck (where the pruritogen injection was administered for the TRAP). […] Recent work from Gao et al. 2019, suggests that activating Tac1 neurons can drive robust scratching behaviors; this suggests that there could be parallel circuits downstream of the CeA than can evoke and inhibit itch evoked scratching.”

2. The parallel already drawn with fear learning, the authors make a beautiful genetic characterization of the neurons that are identified. From fear learning field, a lot has become known about the internal circuitry within the CeA and the (partial) genetic identification of neurons involved: those that trigger the freezing behavior (the somatostatin expressing neurons) and those that inhibit the freezing behavior (PKCdelta neurons, Ciocchi et al., Haubensak et al., Nature 2010) and those that trigger escape (CRF neurons, Fadok et al. Nature, 2018). The authors do mention these different behaviors, but do not discuss the molecular markers that identify them (CRF, PKC δ+, somatostatin). Did the authors find any of the markers that have been used to characterize these neurons in their genetic analyses? How did these markers distribute across the neuronal populations that they identified? What about the neurons identified to be activated by isoflurane? Do they co-segregate with these itch-activated neurons'?

“One other interesting observation from our sequencing dataset is the relative enrichment of Prkcd transcript in FosTRAP+ve vs FosTRAP-ve cells. Prkcd+ve cells in CeA have been shown to be involved in fear and pain processing. It would be interesting to see what role these cells play in pruritic behaviors. It is also possible that the needle stick associated with 4OHT injection could label a small population of CeA neurons involved in fear or pain processing, and this could impact our sequencing dataset to some extent.” This is part of the discussion now.

We have also identified expression of SST and Penk in FosTRAP+ve cells is significantly lower relative to the FosTRAP-ve cells and we hypothesize that these genes could be involved in the suppression of pruritus. We have used (FISH) to visualize mRNA expression and quantify overlap of Penk with itch evoked Fos+ve cells. We found that only 30% of PenK^+^K^+^ve overlap with itch evoked Fos+ve cells. We have not formally quantified expression using FISH for SST and CRF. The Penk dataset is included in the manuscript. Future experiments using scRNA seq should allow us to parse out diversity of cell-types that contribute in pruritic processing.

Our modifications to the manuscript: We have now made changes in results in reference to this, and the following text was added to page 18 line 26 to page 19 line 2 of the revised manuscript:

“One other interesting observation from our sequencing dataset is the relative enrichment of Prkcd transcript in FosTRAP+ve vs FosTRAP-ve cells. […] It is also possible that the needle stick associated with 4OHT injection could label a small population of CeA neurons involved in fear or pain processing, and this could impact our sequencing dataset to some extent.”

3. Finally, the most intriguing question poses itself what would be the role of the scratching and how would scratching affect the sensory sensation of the itching? Generally believed it seems that scratching on the spot of the itch, reduces the itching behavior. In that case would mice that are able to scratch , show a lower activation of the CeA neurons, compared to those that are prevented from scratching the itching spot? And furthermore, is this (negative) feedback loop only existing at the behavioral level outside of the body or is it possible that also at the circuitry level of the CeA a representation of this motor-sensory feedback loop exists (by various types of inhibitory connections that have already been described been other cell-types?). In this same context, it seems odd that neurons that are activated by the itching would be exactly the same neurons that are at the origin of the scratching behavior. An animal may choose not to scratch, for example when facing a dangerous situation where freezing would be of essential for survival (and one can recall a number of Hitchcock-genre videos where actors have faced very similar situations). So somewhere an active inhibition most take place and the CeA, with its many inhibitory neuronal connections between local cell populations might play an important role. However, this would imply different populations for itch and scratch possibly within the CeA. Could the authors discuss this within the context of what is already known in the field?

This is an excellent question. Recent work in the VTA, shows that pruritogen evoked scratching elevates the activity of dopamine neurons and this elevated activity is required for the hedonic aspects of pruritogen evoked scratching. VTA is known to send projections to the CeA, it is possible that these projections encode aversive aspects that we have seen in the CeA. What we could not parse out from our data is if neurons that encode for itch evoked by pruritogen and scratching response to ongoing itch are the same population or distinct CeA sub populations. It is possible, as the reviewer pointed it out, that there are multiple populations that could be driving sensory and motor aspects of itch-scratch behaviors. There is now literature suggesting that scratching on the spot of the itch can suppress neural activity in spinothalamic neurons. In these neurons’ activity elicited by pruritogens can be completely abolished by scratching, suggesting that relief of itch by scratching needs suppression of activity in the spinal cord. There are additional studies now showing that supraspinal projections from the PAG and RVM directly modulate this activity in a state dependent manner. Active inhibition of scratching could take place downstream of CeA when PAG and RVM neurons are engaged in inhibiting ongoing spinal pruritic transmission.

Our modifications to the manuscript: We have now made changes in results in reference to this, and the following text was added to page 17 line 25 to page 18 line 15 of the revised manuscript:

“Recent work in the VTA, shows that pruritogen evoked scratching elevates the activity of dopamine neurons and this elevated activity is required for the hedonic aspects of pruritogen evoked scratching (Su et al., 2019; Yuan et al., 2018). […] Active inhibition of scratching could take place downstream of CeA when PAG and RVM neurons are engaged in inhibiting ongoing spinal pruritic transmission.”